# Facilitating alkaline hydrogen evolution reaction on the hetero-interfaced Ru/RuO$_2$ through Pt single atoms doping

Yiming Zhu[1], Malte Klingenhof[2], Chenlong Gao[1], Toshinari Koketsu [2], Gregor Weiser[2], Yecan Pi[3], Shangheng Liu[4], Lijun Sui[1], Jingrong Hou[1], Jiayi Li[1], Haomin Jiang[5,6], Limin Xu[7,8], Wei-Hsiang Huang [9], Chih-Wen Pao [9], Menghao Yang [1] ✉, Zhiwei Hu [10] ✉, Peter Strasser [2] ✉ & Jiwei Ma [1] ✉

Exploring an active and cost-effective electrocatalyst alternative to carbon-supported platinum nanoparticles for alkaline hydrogen evolution reaction (HER) have remained elusive to date. Here, we report a catalyst based on platinum single atoms (SAs) doped into the hetero-interfaced Ru/RuO$_2$ support (referred to as Pt-Ru/RuO$_2$), which features a low HER overpotential, an excellent stability and a distinctly enhanced cost-based activity compared to commercial Pt/C and Ru/C in 1 M KOH. Advanced physico-chemical characterizations disclose that the sluggish water dissociation is accelerated by RuO$_2$ while Pt SAs and the metallic Ru facilitate the subsequent H* combination. Theoretical calculations correlate with the experimental findings. Furthermore, Pt-Ru/RuO$_2$ only requires 1.90 V to reach 1 A cm$^{-2}$ and delivers a high price activity in the anion exchange membrane water electrolyzer, outperforming the benchmark Pt/C. This research offers a feasible guidance for developing the noble metal-based catalysts with high performance and low cost toward practical H$_2$ production.

At present, seeking a promising energy carrier to replace traditional fossil fuels is more critical than ever to deal with the issues of environmental deterioration and resource depletion[1]. Hydrogen (H$_2$) with the merits of non-pollutant emission and high energy density is regarded as the most suitable alternative. Meanwhile, the electrochemical water splitting powered by renewable energy provides a green and sustainable way for efficient H$_2$ production[2]. Compared with water electrolysis conducted in the acidic medium, alkaline water

electrolysis technology is currently more popular and feasible in practical industry, due to its advantages of robust facility and cheap electrolyzer construction[3,4]. However, the cathodic hydrogen evolution reaction (HER) in alkaline electrolyte involves the prior water dissociation process (Volmer step) and the later hydrogen combination process (Heyrovsky or Tafel step), in which the sluggish kinetics of the Volmer step will severely limit the overall water splitting performance[5,6]. Therefore, it is of great significance to accelerate this

[1]Shanghai Key Laboratory for R&D and Application of Metallic Functional Materials, Institute of New Energy for Vehicles, School of Materials Science and Engineering, Tongji University, 201804 Shanghai, China. [2]Technische Universität Berlin, Department of Chemistry, 10623 Berlin, Germany. [3]School of Chemistry and Chemical Engineering, Yangzhou University, 225002 Jiangsu, China. [4]State Key Laboratory of Physical Chemistry of Solid Surfaces, College of Chemistry and Chemical Engineering, Xiamen University, 361005 Xiamen, China. [5]Baosteel Central Research Institute, Baoshan Iron & Steel Co., Ltd., 201999 Shanghai, China. [6]State Key Laboratory of Development and Application Technology of Automotive Steels, Baosteel, 201900 Shanghai, China. [7]Baowu Aluminum Technical Center, Baosteel Central Research Institute, Baoshan Iron & Steel Co., Ltd., 201999 Shanghai, China. [8]Shanghai Engineering Research Center of Metals for Lightweight Transportation, 201999 Shanghai, China. [9]National Synchrotron Radiation Research Center, Hsinchu 30076, Taiwan. [10]Max Planck Institute for Chemical Physics of Solids, Nöthnitzer Strasse 40, 01187 Dresden, Germany. ✉e-mail: menghaoyoung@tongji.edu.cn; zhiwei.hu@cpfs.mpg.de; pstrasser@tu-berlin.de; jiwei.ma@tongji.edu.cn

restrictive barrier by discovering an efficient electrocatalyst towards alkaline HER.

Although the platinum (Pt) nanomaterials with optimal hydrogen binding energy are widely acknowledged as the most active catalysts toward HER, their catalytic activities in the alkaline medium are approximately two orders of magnitude lower than those in the acidic medium. This can be mainly attributed to the shortage of water dissociation ability on Pt atoms and thus leads to the insufficient supply of protons for $H_2$ generation[7–9]. Conversely, metal oxides/hydroxides have been regarded as the ideal candidates for oxygen evolution reaction (OER) because of their good capabilities for cleaving the strong H-OH bond, whereas they are inefficient for propelling the hydrogen generation process in HER[10–12]. Generally, one well-designed electrocatalyst for activating HER in the alkaline electrolyte should provide a platform with different active sites for water dissociation and hydrogen adsorption steps to take place, respectively. Enlightened by this guideline, many efforts have recently been devoted to construct the composite engineering by integrating Pt with metal oxide/hydroxides to synergistically accelerate the alkaline HER. For instance, Subbaraman et al. reported that the kinetic barrier of the Volmer step on Pt nanomaterials can be availably promoted by the decorated nickel hydroxide ($Ni(OH)_2$) clusters, yielding reduced HER overpotentials in KOH solution[13]. Wang and coworkers also successfully prepared the dense Pt nanoparticles immobilized oxygen vacancy-rich $NiO_X$ heterojunctions as an alkaline HER electrocatalyst with an enhanced performance[14]. In addition, He et al. demonstrated that the water dissociation and hydrogen generation are speeded up on Pt nanomaterials incorporated Ni–iron layered double hydroxides (Ni-Fe LDH)[15]. Despite the progress that has been achieved, the low abundance and high cost of Pt severely restrict its large-scale applications in the practical industry[16,17]. In this regard, it is necessary to take the high HER activity and minimized Pt usage amount into consideration simultaneously when preparing the Pt-based hybrid electrocatalysts. Recently, the highly dispersed single-atom catalysts (SACs) are considered as the most promising candidates for a range of electrochemical reactions because of their distinct merits such as maximized atom-utilization efficiency, well-defined active sites and significantly reduced preparation cost[18–23]. Therefore, the combination of Pt single atoms and metal oxide/hydroxide supports shows brightening prospect for balancing the tip between low cost and high performance in catalytic reactions. Meanwhile, as one of the cheapest noble metals, the cost-effective ruthenium (Ru) and its derivatives possess the strongest ability to prompt the water dissociation step, which can be employed as the supports for Pt single atoms to fully unlock the potential for catalyzing the alkaline HER, while it has no related report to date[24–27].

In this work, we report the synthesis, characterization and electrocatalysis of an active and cost-efficient HER electrocatalyst for the use at cathode of alkaline water electrolyzers. Our HER catalyst design involves individual Pt single metal atoms doped into the heterointerfaced Ru/RuO$_2$ supports (referred to as Pt–Ru/RuO$_2$). More specifically, this catalyst with low-Pt content requires very low overpotentials of merely 18 mV and 63 mV at the current densities of 10 mA cm$^{-2}$ and 250 mA cm$^{-2}$, respectively, outperforming most recently reported noble metal and noble metal-free electrocatalysts. The noble metal cost-based catalytic activity of Pt–Ru/RuO$_2$ reaches a factor of more than 16 compared to those of commercial Pt/C and Ru/ C at the overpotential of 63 mV. Notably, Pt–Ru/RuO$_2$ delivers a long-term stability over 100 h with negligible activity loss. Corresponding mechanism investigated by various experiments and operando characterizations validate that all isolated Pt atoms, Ru and RuO$_2$ in the support play vital roles in activating the alkaline HER. Concretely, the sluggish water dissociation step is accelerated by the RuO$_2$, the following hydrogen combination step is primarily promoted on Pt single atoms and also partially contributed by Ru, synergistically resulting in a remarkable activity of Pt–Ru/RuO$_2$. Density functional theory (DFT)

calculations further determine the specific functions of active sites in Pt–Ru/RuO$_2$, which corroborate the experimental discoveries. Moreover, the integration of Pt–Ru/RuO$_2$ and NiFe LDH in the anion exchange membrane water electrolyzer (AEMWE) only requires 1.90 V to reach the large current density of 1 A cm$^{-2}$ and exhibits a high price activity of 247.1 A dollar$^{-1}$ at 2.1 V, much superior to the Pt/C-based counterpart. This research finding stimulates our interest to pursue the balance between cost-effectiveness and high activity when exploring noble metal-based electrocatalysts toward practical $H_2$ production.

## Results

### Structural characterizations of Pt–Ru/RuO$_2$

To synthesize the Pt–Ru/RuO$_2$, the RuO$_2$ enriched with grain boundaries was obtained first by immersing the RuCl$_3$ precursor in the molten NaNO$_3$ (Supplementary Figs. 1, 2). Then Pt species were absorbed on the synesized RuO$_2$ followed by calcination in Ar, in which process the Pt single atoms were incorporated and RuO$_2$ was slightly reduced to interfaced Ru/RuO$_2$ (see Methods for details). As shown in Fig. 1a, the flat-spread and dense nanostructure of Pt–Ru/RuO$_2$ was initially characterized by transmission electron microscopy (TEM). A typical high-resolution TEM (HR-TEM) image indicates that the Pt–Ru/RuO$_2$ consists of independently distributed crystal regions belong to Ru and RuO$_2$, with evident grain boundaries among them. To be specific, Ru and RuO$_2$ crystal regions reveal the lattice spacings of 2.07 Å and 3.18 Å, which can be ascribed to the Ru (101) and RuO$_2$ (110) facets, respectively (Fig. 1b)[28]. Generally, the existence of grain boundary suggests the close contact between two phases, favorable for inducing efficient interaction such as electron transfer in electrochemical reactions[29,30]. In addition, Ru/ RuO$_2$ was prepared for comparison. As depicted in Supplementary Fig. 3, Ru/RuO$_2$ shows a similar morphology and interfaced structure as the Pt–Ru/RuO$_2$. To reveal the detailed crystal structures of Pt–Ru/RuO$_2$ and Ru/RuO$_2$, X-ray diffraction (XRD) was employed and corresponding results suggest that two phases attributed to Ru (JCPDS No. 06-0663) and RuO$_2$ (JCPDS No. 43-1027) are coexisted in these patterns, which is in keep with the finding of HR-TEM. Moreover, no diffraction peak related to Pt can be found in these patterns. Rietveld refinements were further performed on the primary peaks of those XRD patterns to calculate the relative content of Ru and RuO$_2$. As a result, the Ru molar ratio is calculated to be 15.03% in Ru/RuO$_2$, and a slightly more Ru (25.18%) appears on Pt–Ru/RuO$_2$ because of the introduction of Pt species (Supplementary Figs. 4, 5 and Supplementary Tables 1, 2)[31]. Afterwards, scanning electron microscopy-energy dispersive spectroscopy (SEM-EDS) and inductively coupled plasma-optical emission spectrometry (ICP-OES) determine that the atomic ratio of Pt is 1.36% in Pt–Ru/RuO$_2$ (Supplementary Fig. 6).

In order to disclose the atomic distribution of Pt species, aberration-corrected high-angle annular dark field-scanning TEM (AC HAADF-STEM) was employed on Pt–Ru/RuO$_2$. As revealed in Fig. 1c, d, two distinct crystal regions with clear atomic arrangements named as Pos 1 and Pos 2 can be assigned to the RuO$_2$ (110) and Ru (101) facets, respectively. Interestingly, owing to the higher Z-contrast of Pt against other elements, the isolated bright spots correspond to Pt atoms are directly visualized and homogenously dispersed on both Ru and RuO$_2$ regions, demonstrating that Pt species mainly exist as the single atoms on the Ru/RuO$_2$ supports[32,33]. This is further confirmed by the atomic intensity profiles along the dashed rectangles collected in Fig. 1d, from which the isolated Pt atoms with higher intensities are observed (Fig. 1e). Additionally, corresponding elemental mappings indicate that Pt atoms are homogeneously distributed over the entire Pt–Ru/ RuO$_2$ (Fig. 1f and Supplementary Fig. 7). Afterwards, the surface electronic structures and chemical compositions were analyzed by X-ray photoelectron spectroscopy (XPS). The survey XPS spectrum initially proves the existence of Pt elements in Pt–Ru/RuO$_2$ through displaying an obvious Pt 4$f$ signal (Supplementary Fig. 8a). In addition, both Ru 3$p$ XPS spectra on Pt–Ru/RuO$_2$ and Ru/RuO$_2$ can be deconvoluted into

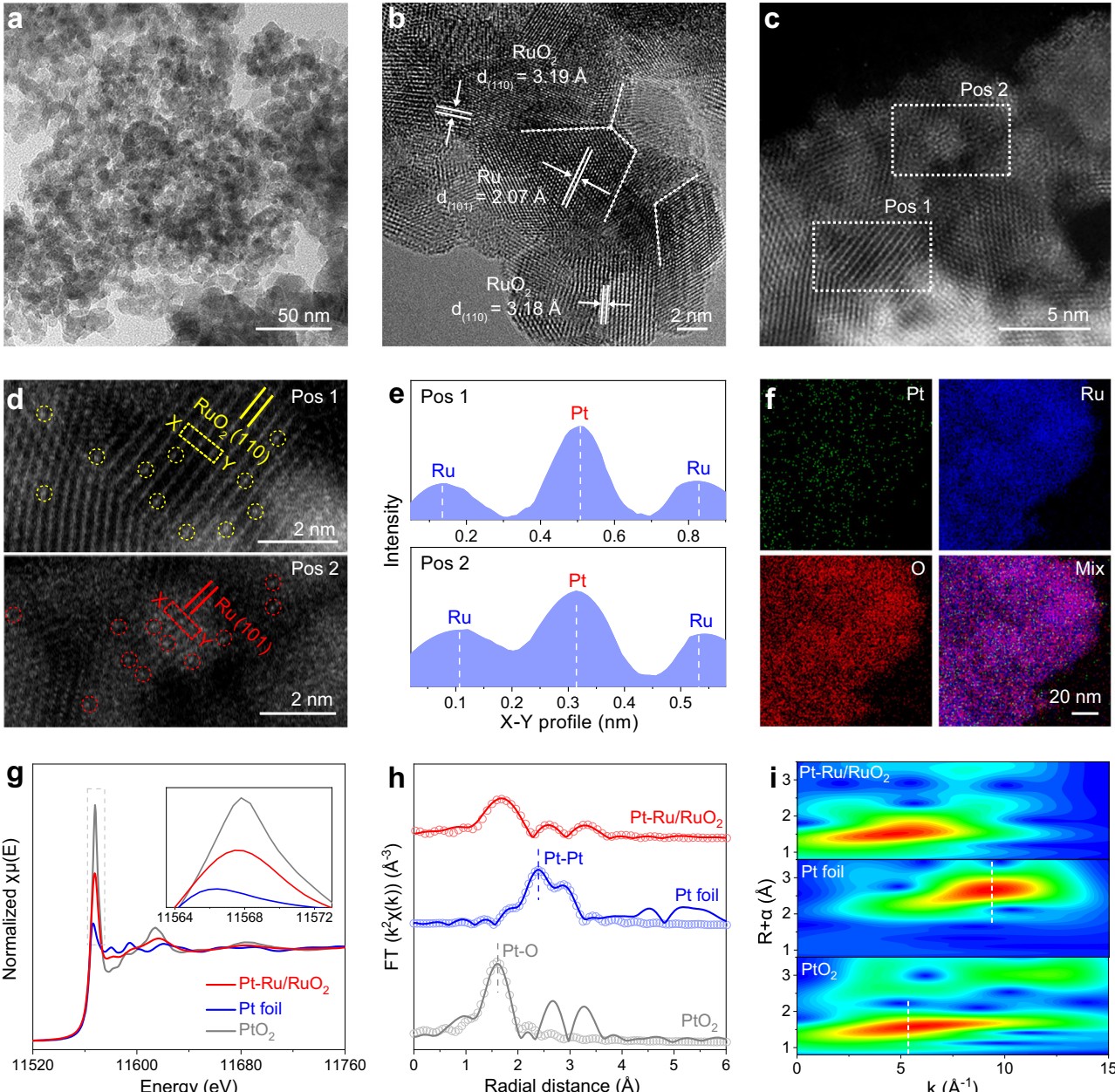

**Fig. 1 | Structural characterizations of Pt–Ru/RuO₂. a** TEM, **b** HR-TEM and (**c**) AC HAADF-STEM images of Pt–Ru/RuO₂. **d** The enlarged areas of Pos 1 and Pos 2 in (**c**), with the Pt single atoms marked in circles, respectively. **e** Atomic intensity profiles along the dashed rectangles in (**d**). **f** Corresponding elemental mappings of Pt–Ru/RuO₂. **g** The normalized Pt $L_3$-edge XANES spectra of Pt–Ru/RuO₂, Pt foil and PtO₂. **h** Pt $L_3$-edge EXAFS spectra and corresponding fitting curves, and (**i**) wavelet-transformed $k^2$-weighted EXAFS spectra of Pt–Ru/RuO₂, Pt foil and PtO₂.

doublets of Ru⁰ and Ru⁴⁺, implying the coexistence of metallic Ru and RuO₂[34]. After the introduction of Pt, the ratio of Ru⁰/Ru⁴⁺ is increased from 0.16 to 0.28, which is in good accordance with the XRD results (Supplementary Fig. 8b)[31]. Supplementary Fig. 8c shows the O 1s spectra on these samples, where the peaks at the binding energy of 529.3 eV, 530.9 eV and 532.5 eV correspond to lattice O, absorbed OH⁻/O₂ and absorbed H₂O, respectively[35]. Meanwhile, Pt 4f XPS spectrum indicates that the commercial Pt/C primarily contains the peak ascribed to metallic Pt. It is worth noting that both metallic Pt and oxidized Pt are found on Pt–Ru/RuO₂. The single-atom alloy is formed when Pt single atoms are homogeneously dispersed in the region of Ru, in which Pt features the metallic characteristics and thus has a valence state of zero[36–38]. Meanwhile, the oxidized Pt is primarily originated from the Pt single atoms coordinated by O atoms on the RuO₂ (Supplementary Fig. 8d).

Subsequently, the detailed chemical states and the local atomic structures were determined by synchrotron-radiation-based X-ray absorption spectroscopy (XAS). The oxidation state of Pt in Pt–Ru/RuO₂ was first detected by X-ray absorption near-edge structure (XANES) along with standard Pt foil and PtO₂ as the references. It is acknowledged that the white line position at the 5d $L_3$-edge XANES is highly sensitive to the valence state of 5d element[39]. As illustrated in Fig. 1g, the white-line energy position of Pt–Ru/RuO₂ is located between Pt foil and PtO₂, implying the oxidized state of Pt species is between 0 and +4 in Pt–Ru/RuO₂[40]. Moreover, the extended X-ray absorption fine structure (EXAFS) is a powerful tool for discovering the atomic coordination environment. It can be clearly observed that Pt–Ru/RuO₂ exhibits a dominant peak at 1.67 Å in the R-space of the Pt $L_3$-edge EXAFS spectrum, corresponding to Pt–O bond. Additionally, no discernible peak related to Pt–Pt bond can be distinguished in

Pt–Ru/RuO$_2$, evidencing the absence of Pt nanoparticles or clusters (Fig. 1h). This was further consolidated by the k-space of wavelet transform (WT)-EXAFS, in which Pt–Ru/RuO$_2$ shows a maximum scattering at $R = 1.53$ Å and $k = 4.90$ Å$^{-1}$, close to the Pt–O scattering in PtO$_2$. Also, no scattering corresponds to Pt–Pt bond is found in comparison with Pt foil ($R = 2.68$ Å and $k = 9.35$ Å$^{-1}$), proving the lack of agglomerated Pt in Pt–Ru/RuO$_2$ (Fig. 1i)[41]. Furthermore, a model-based EXAFS fit was performed on Pt–Ru/RuO$_2$, PtO$_2$ and Pt foil. Results confirm that the main peak located at 1.67 Å is Pt–O bond, other peaks at 2.59 Å and 3.29 Å are derived from Pt–Ru and Pt–O–Ru moieties, which suggests that the isolated Pt atoms in Pt–Ru/RuO$_2$ are coordinated by surrounded O atoms from RuO$_2$ regions and surrounded Ru atoms from Ru regions, respectively (Supplementary Fig. 9 and Supplementary Table 3)[32,42]. Therefore, combining the detailed AC HAADF-STEM and EXAFS analysis, we conclude that the Pt species are existed as single atoms and homogenously dispersed on the interfaced Ru/RuO$_2$ supports. In another case, the Ru valence states on Pt–Ru/RuO$_2$ were also evaluated by Ru $K$-edge XANES spectra together with Ru foil, RuO$_2$ and Sr$_2$GdRuO$_6$ for comparison. As shown in Supplementary Fig. 10a, the absorption edge on Pt–Ru/RuO$_2$ displays a lower energy shift compared with Ru/RuO$_2$, indicating the lower valence state of the former, which is due to the higher fraction of metallic Ru in Pt–Ru/RuO$_2$ according to XRD and XPS results. Through establishing the standard curves between the Ru valence states and the energy of white-line absorption edges, the average oxidation states of Ru on Pt–Ru/RuO$_2$ and Ru/RuO$_2$ are determined as +2.8 and +3.3, respectively (Supplementary Fig. 10b). Besides, both the Ru $K$-edge EXAFS spectra of Pt–Ru/RuO$_2$ and Ru/RuO$_2$ show the identical peaks as RuO$_2$ except the subtle peaks attributed to Ru–Ru bond present at 2.30 Å, which signifies the existence of reduced Ru and consistent with abovementioned characterizations (Supplementary Fig. 10c).

## Electrochemical performance of Pt–Ru/RuO$_2$

The electrochemical performances of those prepared electrocatalysts toward HER were evaluated in 1 M KOH electrolyte involving a three-electrode system. Similar tests were also carried out on the commercial Pt/C, Ru/C and RuO$_2$ for explicit comparison (Supplementary Fig. 11). All measured potentials in this study were carefully calibrated to the reversible hydrogen electrode (RHE) scale (Supplementary Fig. 12). The actual resistance values for all studied catalysts were detected in 1 M KOH solution (Supplementary Fig. 13a) and the non-iR corrected linear sweep voltammetry (LSV) curves are shown in Supplementary Fig. 13b. As displayed in Fig. 2a, the LSV curves with 95% iR corrections show the activity trend of Pt–Ru/RuO$_2$ > Pt/C > Ru/C > Ru/RuO$_2$ > RuO$_2$ at low potentials. It is noteworthy that Pt–Ru/RuO$_2$ and Ru/RuO$_2$ exhibit an accelerated rise in activity relative to other catalysts as the potential increases, on account of the speedy HER rates. To be specific, Pt–Ru/RuO$_2$ delivers an overpotential of 18 mV at the current density of 10 mA cm$^{-2}$, which is significantly lower than those of Pt/C (45 mV), Ru/C (49 mV), Ru/RuO$_2$ (112 mV), and RuO$_2$ (110 mV). Impressively, at a higher current density of 250 mA cm$^{-2}$, Pt–Ru/RuO$_2$ still possesses the smallest overpotential (63 mV) among all studied electrocatalysts, indicating its remarkable alkaline HER activity (Fig. 2b). In addition, the synthesized RuO$_2$ shows the almost same HER performance as the commercial RuO$_2$ (Supplementary Fig. 14). Moreover, Fig. 2c illustrates the Tafel slopes based on the LSV curves and the corresponding values are 18.5 mV dec$^{-1}$, 50.7 mV dec$^{-1}$, 64.0 mV dec$^{-1}$, 55.0 mV dec$^{-1}$, and 120.5 mV dec$^{-1}$ for Pt–Ru/RuO$_2$, Pt/C, Ru/C, Ru/RuO$_2$, and RuO$_2$, respectively. The lowest Tafel slope of Pt–Ru/RuO$_2$ indicates the most effective facilitation of the hydrogen evolution kinetic[25]. In addition, such a low Tafel slope value on Pt–Ru/RuO$_2$ also indicates that it follows the Volmer-Tafel mechanism in alkaline HER after the incorporation of Pt single atoms with high hydrogen coverage and binding ability[43,44]. In order to further highlight the advantages of overpotential and Tafel slope, a synopsis of performances on recently reported Pt-based and Ru-based HER electrocatalysts in alkaline medium is listed in Fig. 2d and Supplementary Table 4, in which Pt–Ru/RuO$_2$ is much superior to others. Additionally, the double-layer capacitance ($C_{dl}$) measurements derived from a series of cyclic voltammetry (CV) with various scanning rates were employed to investigate the intrinsic activities of these samples. As shown in Supplementary Fig. 15, the larger $C_{dl}$ value of Pt–Ru/RuO$_2$ as compared to Ru/RuO$_2$ indicates the introduced Pt single atoms lead to a larger electrochemical surface area (ECSA) and more exposed active sites[35]. Accordingly, by normalizing the HER current to ECSA at the overpotential of 63 mV, Pt–Ru/RuO$_2$ possesses the highest specific activity (0.89 mA cm$^{-2}_{ECSA}$), almost 5 times higher than that of benchmark Pt/C (0.16 mA cm$^{-2}_{ECSA}$) (Supplementary Fig. 16). In addition, due to the efficient hydrogen adsorption ability of Pt and Ru, the hydrogen underpotential deposition (H$_{upd}$) method was employed to measure the ECSA of Pt/C, Pt–Ru/RuO$_2$ and Ru/RuO$_2$. As demonstrated in Supplementary Fig. 17, after the incorporation of Pt single atoms, Pt–Ru/RuO$_2$ exhibits a similar ECSA value with Pt/C, which is much larger than the Ru/RuO$_2$ and RuO$_2$. Consequently, Pt–Ru/RuO$_2$ also shows the highest specific activity among studied catalysts. Furthermore, the ECSA values and specific activities calculated by the CO stripping method are in highly accord with those derived from the H$_{upd}$ method (Supplementary Fig. 18). BET-normalized activities were also investigated to exclude the morphological influences on evaluating performances, it can be observed in Supplementary Fig. 19 that Pt–Ru/RuO$_2$ still outperforms Ru/RuO$_2$. Mass activity is another essential factor for the evaluation of an electrocatalyst system owing to the direct relationship with the cost. Supplementary Fig. 20 plotted the mass activities (normalized to the loading mass of Pt and Ru) of these studied materials at the overpotential of 63 mV. As a result, Pt–Ru/RuO$_2$ delivers a mass activity of 2227.3 A g$^{-1}_{Pt+Ru}$, which is almost 14 and 17 times higher than those of the commercial Pt/C (164.4 A g$^{-1}_{Pt}$) and Ru/C (134.8 A g$^{-1}_{Ru}$), showing its greater merit of cost-effectiveness. Surprisingly, the calculated price activity (normalized to the price of Pt and Ru) of Pt–Ru/RuO$_2$ (116.7 A dollar$^{-1}$) can also reach 21.8 and 16.3 folds greater than those of the commercial Pt/C (5.34 A dollar$^{-1}$) and Ru/C (7.17 A dollar$^{-1}$), respectively, further exhibiting its economic efficiency and brighten potential toward industrial applications (Fig. 2e). Moreover, the two other Pt–Ru/RuO$_2$ catalysts with various amounts of Pt dopant were prepared by halving and doubling the inputs of Pt precursor, their Pt atomic ratios are determined to be 0.92% and 3.06% by ICP, respectively. LSV results in Supplementary Fig. 21 show that after doping 0.92% Pt, the HER activity of Pt–Ru/RuO$_2$ is obviously enhanced compared to the Ru/RuO$_2$ due to the incorporation of highly active Pt single sites. However, the HER activity of Pt–Ru/RuO$_2$ decreases with further increasing the amount of Pt dopant from 1.36% to 3.06%, which may be attributed to the formation of aggregated Pt clusters, suggesting that the optimal amount of Pt dopant in Pt–Ru/RuO$_2$ toward alkaline HER is 1.36%. In addition, the Nyquist plots are obtained by establishing the equivalent circuit in electrochemical impedance spectroscopy (EIS) measurements followed by fitting processes (Supplementary Fig. 22 and Supplementary Table 5). As depicted in Fig. 2f, compared to Pt/C (10.4 Ω), Ru/C (21.7 Ω), Ru/RuO$_2$ (121.2 Ω), and RuO$_2$ (196.8 Ω), Pt–Ru/RuO$_2$ possesses the smallest charge transfer resistance ($R_{ct}$) of 0.42 Ω, implying the expedited charge transfer rate and thus faster HER kinetics.

Apart from the activity, durability is also critical to assess an HER electrocatalyst, especially when targeted for industrial hydrogen production applications. The stability tests on Pt–Ru/RuO$_2$ and commercial Pt/C were carried out by chronopotentiometry at various current densities. As presented in Fig. 2g, it is seen that the activity of Pt–Ru/RuO$_2$ at 10 mA cm$^{-2}$ remains almost steady without an obvious decay during the constant operation of 100 h, whereas the overpotential of Pt/C exhibits a rapid 55 mV increase after the continuous 100 h test. By determining the dissolved ion concentrations in the electrolytes after a

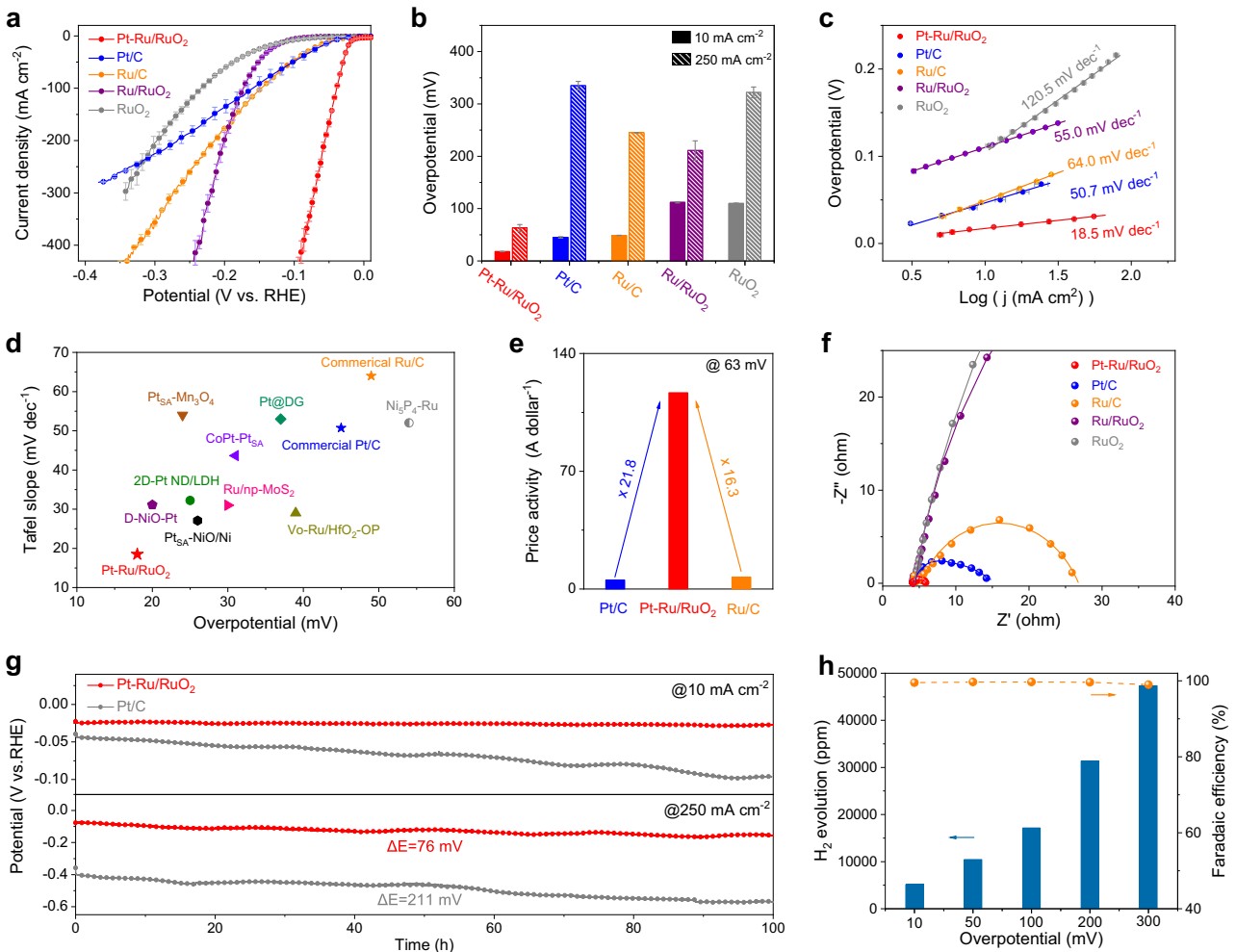

**Fig. 2 | Alkaline HER performances of Pt–Ru/RuO₂. a** Polarization curves with 95% iR corrections and (**b**) corresponding overpotentials of Pt–Ru/RuO₂, Ru/RuO₂, RuO₂, Pt/C and Ru/C at the current densities of 10 mA cm⁻² and 250 mA cm⁻² in 1 M KOH. **c** Tafel plots derived from the polarization curves in (**a**). **d** Comparison of overpotentials and Tafel plots in recently reported Pt, Ru-based HER electrocatalysts. **e** Comparison of the price activities on Pt–Ru/RuO₂, Pt/C and Ru/C at the overpotential of 63 mV. The prices of Pt and Ru are sourced from U.S. Geological Survey (Date: Average price of platinum-group metals in January-March 2022). **f** EIS Nyquist plots and fitting curves of these samples recorded at −50 mV. **g** Chronopotentiometric measurements of Pt–Ru/RuO₂ and Pt/C at current densities of 10 mA cm⁻² and 250 mA cm⁻², respectively, carbon paper was used as the catalyst support. **h** Generated H₂ amount and corresponding Faradaic efficiency over a range of overpotentials on Pt–Ru/RuO₂. Note: error bars represent the standard deviation of three independent measurements.

long-term reaction, the deactivated reason for Pt/C is attributed to the severe Pt ion leaching. In contrast, slight dissolutions of both Pt and Ru ions are occurred on Pt–Ru/RuO₂, resulting in the enhanced HER stability (Supplementary Fig. 23). Post-reaction characterizations of TEM and HR-TEM confirm that the overall morphology and the interfaced structure of Pt–Ru/RuO₂ are well preserved except for slight agglomerations. In addition, elemental mappings indicate that Pt atoms are still homogeneously dispersed on the Ru/RuO₂ supports and no obvious crystal structure changes are found from the XRD pattern after the stability test at 10 mA cm⁻² (Supplementary Fig. 24). Meanwhile the Pt 4$f$ and Ru 3$p$ XPS spectra suggest the decreased valence states of Pt and Ru on Pt–Ru/RuO₂, which is common for HER catalysts after long-term operation under the reduction potentials (Supplementary Fig. 25). Moreover, the stabilities tests were also conducted at a higher current density of 250 mA cm⁻². Significantly, compared with the dramatic increase of overpotential on Pt/C (211 mV), only a 76 mV increase of overpotential is shown on Pt–Ru/RuO₂ after 100 h measurements.

The above measurements demonstrate the excellent HER stability of Pt–Ru/RuO₂ under the conditions of strong alkaline and high voltage. The repeated CV scanning experiments further highlight that the

polarization curves of Pt–Ru/RuO₂ remain almost identical before and after 5000 CVs, whereas other samples display obvious activity losses (Supplementary Fig. 26). Besides, in order to avoid the interference of the side reactions and thus the overestimation of the actual HER activity, the generated H₂ amounts on Pt–Ru/RuO₂ over a wide range of overpotentials are monitored by gas chromatography and corresponding Faradaic efficiencies (FE) are calculated (Supplementary Fig. 27)[45]. As illustrated in Fig. 2h, with the increasing of applied potentials, the H₂ evolution amount shows a rapid upward tendency, meanwhile the associated FEs keep nearly 100%. This signifies that the majority of electrochemical energy is employed to drive HER and almost no side reactions take place on Pt–Ru/RuO₂. Overall, Pt–Ru/RuO₂ features the advantages of high activity, cost-effectiveness and remarkable stability toward alkaline HER, demonstrating its promising prospect to be deployed in the practical hydrogen production applications.

### Mechanistic insights into alkaline HER
Considering the well-designed Pt–Ru/RuO₂ has been demonstrated as a high-performance alkaline HER catalyst, the contributions of various active sites were explored. Poisoning experiments were initially carried

out via potassium thiocyanate (KSCN) as the deactivator to block the contact between active sites and reaction intermediates[46]. As shown in Supplementary Fig. 28, the significant decline of activity on Pt–Ru/RuO$_2$ after the addition of SCN$^-$, implying the essential role of Pt single atom toward activating HER. Besides, compared with the RuO$_2$, the obvious activity loss on Ru/RuO$_2$ indicates that Ru also contributes to alkaline HER (Supplementary Fig. 29). To further get a deep insight into the improved mechanism of Pt single atoms, the operando EIS measurements were performed during HER. The Nyquist plots were stimulated by a double-parallel equivalent circuit model, in which the pseudo-capacitance ($C_\varphi$) and hydrogen absorption resistance ($R_3$) are adopted for tracking the absorbed hydrogen (H*) behavior on the catalyst surface[47,48]. Notably, the Tafel slope (log $R_3$ vs. potentials) of Pt–Ru/RuO$_2$ exhibits a markedly decreased value of 12.9 mV dec$^{-1}$ in comparison to Ru/RuO$_2$ (36.2 mV dec$^{-1}$) and RuO$_2$ (40.5 mV dec$^{-1}$), suggesting that H* adsorption kinetics is facilitated by the introduced Pt single atoms. In addition, the integration of $C_\varphi$ and potential determines the H* adsorption charge ($Q_H$), referring to the H* absorption concentration. It is seen that Pt–Ru/RuO$_2$ possesses a much higher $Q_H$ than that of RuO$_2$, supporting that the Pt single atoms contribute to the improved H* coverage on the surface. Moreover, the relative smaller Tafel slope and higher $Q_H$ of Ru/RuO$_2$ than RuO$_2$ also indicates the reduced Ru can simultaneously improve the H* adsorption (Supplementary Fig. 30 and Supplementary Table 6). These findings are also evidenced by H$_{upd}$ results in Supplementary Fig. 17, while the introduced Pt single atoms and Ru can increase the hydrogen adsorption area, indicating more H atoms are absorbed and leading to efficient H$_2$ combination process. It is worth mentioning that although Pt and Ru are hydrogen-binding active sites, the former are more essential and indispensable considering the relatively larger concentration of adsorbed H* and remarkable enhancement of HER activity induced by Pt single atoms. On the other hand, it is well known that the lower the CO stripping potential, the better the water dissociation ability. As seen in Supplementary Fig. 18, the CO stripping peak potentials on Pt–Ru/RuO$_2$ and Ru/RuO$_2$ are obviously lower than that on Pt/C (0.694 V), confirming the excellent water dissociation ability is not attributed to the doped Pt. Supplementary Fig. 31 further depicts the kinetic isotope effects (KIE) of Pt–Ru/RuO$_2$ and Pt/C in H$_2$O and D$_2$O electrolytes. The smaller KIE value of Pt–Ru/RuO$_2$ implies the easier HO-H bond cleavage, which is beneficial for the water dissociation process[49]. In order to comprehensively understand the enhanced water dissociation ability on Pt–Ru/RuO$_2$, operando Raman spectroscopy was conducted at various potentials (Supplementary Fig. 32). As depicted in Fig. 3a–c, one broad peak at the Raman shift of around 3500 cm$^{-1}$ can be observed in the spectra, which can be ascribed to the water at the catalyst-electrolyte interface. The two carbon signals of D and G bands at around 1400 and 1600 cm$^{-1}$ are derived from the glassy carbon plate (Supplementary Fig. 33a). In addition, in order to confirm that the Raman laser was focused on the surface of electrocatalyst, the operando Raman tests under various potentials in 1 M KOH solution were conducted on the glassy carbon plate without electrocatalyst. It can be clearly seen that almost no interfacial water signals (3500 cm$^{-1}$) can be found in Supplementary Fig. 33b, which is due to the inert property of the glassy carbon plate towards the catalytic reaction. Furthermore, this interfacial water peak can be further fitted into three peaks nearly located at 3225 cm$^{-1}$ ($\upsilon$1), 3450 cm$^{-1}$ ($\upsilon$2) and 3615 cm$^{-1}$ ($\upsilon$3), respectively (Supplementary Figs. 34–36). Generally, $\upsilon$1 and $\upsilon$2 are assigned to the tetrahedral and trihedral coordinated water molecules that participate in the HER, while $\upsilon$3 belongs to the dangling O-H bond of the interfacial water molecule that is inactive toward HER[50]. It is significant to observe that the $\upsilon$3 peak proportions on both Pt–Ru/RuO$_2$ and Ru/RuO$_2$ display a dramatically faster downward trend than that of Pt/C with the increased HER potentials, indicating the more efficient capability on Pt–Ru/RuO$_2$ and Ru/RuO$_2$ for cleaving water bond. In addition, the decline rate of $\upsilon$3 peak proportion on Pt–Ru/

RuO$_2$ is almost the same as that on Ru/RuO$_2$, which suggests that Pt single atoms have no contribution on water dissociation step (Fig. 3d and Supplementary Table 7)[51]. As shown in Supplementary Figs. 37–39, the operando Raman tests were further conducted on Ru and RuO$_2$. After carefully fitting these spectra, it can be obviously observed that RuO$_2$ exhibits a faster downward trend of $\upsilon$3 peak proportion than that on Ru in HER process, confirming its greater capability for cleaving the water bond. Significantly, RuO$_2$ exhibits a similar downward trend as the Ru/RuO$_2$ during more HER cathodic potentials, which validates that RuO$_2$ is the main active site for water activation and dissociation step in Ru/RuO$_2$ instead of Ru. Furthermore, it is calculated that the decline rate of $\upsilon$3 peak proportion on RuO$_2$ (52.4%) is slightly slower than that on Ru/RuO$_2$ (57.5%), which may be attributed to the electronic interaction between Ru and RuO$_2$ in Ru/RuO$_2$ (Supplementary Fig. 40). Based on above investigations, it can be experimentally concluded that RuO$_2$ is primarily responsible for the water dissociation step while both Pt single atoms and Ru facilitate the followed hydrogen combination step, collectively leading to the remarkable alkaline HER activity of Pt–Ru/RuO$_2$, and this is further validated by DFT calculations.

Furthermore, the operando XAS techniques were accepted for monitoring the dynamic changes of electronic structures and local atomic environments on Pt–Ru/RuO$_2$ in the HER process (Supplementary Fig. 41). During the measurements, the working potential was reduced from the open circuit potential (OCP) to −0.2 V vs. RHE. As shown in Fig. 3e, once the voltage of −0.2 V was imposed, corresponding Pt $L_3$-edge white line peak position exhibits an obvious negative shift, manifesting that the Pt single atoms are reduced to lower oxidation states during HER[52,53]. In addition, the real-time change of Ru chemical state was also checked by the operando Ru $K$-edge XANES. Figure 3f illustrates that the absorption edge negatively shifts to lower energy position under the HER working potential of −0.2 V, suggesting the decreased Ru valence state of Pt–Ru/RuO$_2$ during HER (Fig. 3g). As a consequence, the continuously decreased valence states of Pt and Ru during catalytic reactions discovered by operando XANES results elucidate that both Pt single atoms and the Ru/RuO$_2$ support in Pt–Ru/RuO$_2$ are functionalized and jointly participate in activating the alkaline HER[54]. In order to further confirm that both Pt and Ru are active sites during HER, the other low HER operating potentials of −0.018 V (at which the current density is 10 mA cm$^{-2}$) and −0.040 V were applied on Pt–Ru/RuO$_2$ to conduct the operando Pt $L_3$-edge and Ru $K$-edge XANES. Similarly, both white line peak position of Pt and adsorption edge of Ru continuously shift to lower energy with the negatively increased potentials, indicating their decreased valence states in the HER process (Supplementary Fig. 42). Thus, the potential-dependent operando XANES results further verify that both Pt and Ru elements in Pt–Ru/RuO$_2$ are active sites toward the alkaline HER. Moreover, as shown in Fig. 3h, no obvious Pt–Pt bond can be observed in the operando Pt $L_3$-edge EXAFS spectra during HER, indicating that the Pt atoms on Pt–Ru/RuO$_2$ remain isolated dispersion throughout the reaction. Notably, it is seen that the main Pt-O bond shifts to a more positive position at −0.2 V, which may due to the local structure relaxation of Pt caused by the absorbed H* at HER working potential[55]. This is also a strong evidence consistent with experimental results to confirm that Pt single atoms principally take charge of promoting the H* atom adsorption and enhancing the Tafel step in alkaline HER. Operando Ru $K$-edge EXAFS spectra further exhibit that the intensity of Ru–O bond decreases while the intensity of Ru–Ru bond increases with the applied reaction potentials, manifesting the reduction of the Ru/RuO$_2$ supports during HER (Fig. 3i)[56].

## Theoretical investigations of Pt–Ru/RuO$_2$

To further elucidate the mechanism of the excellent activity on Pt–Ru/RuO$_2$ toward alkaline HER, a theoretical study was performed based on the density functional theory (DFT) calculations. The atomic models of

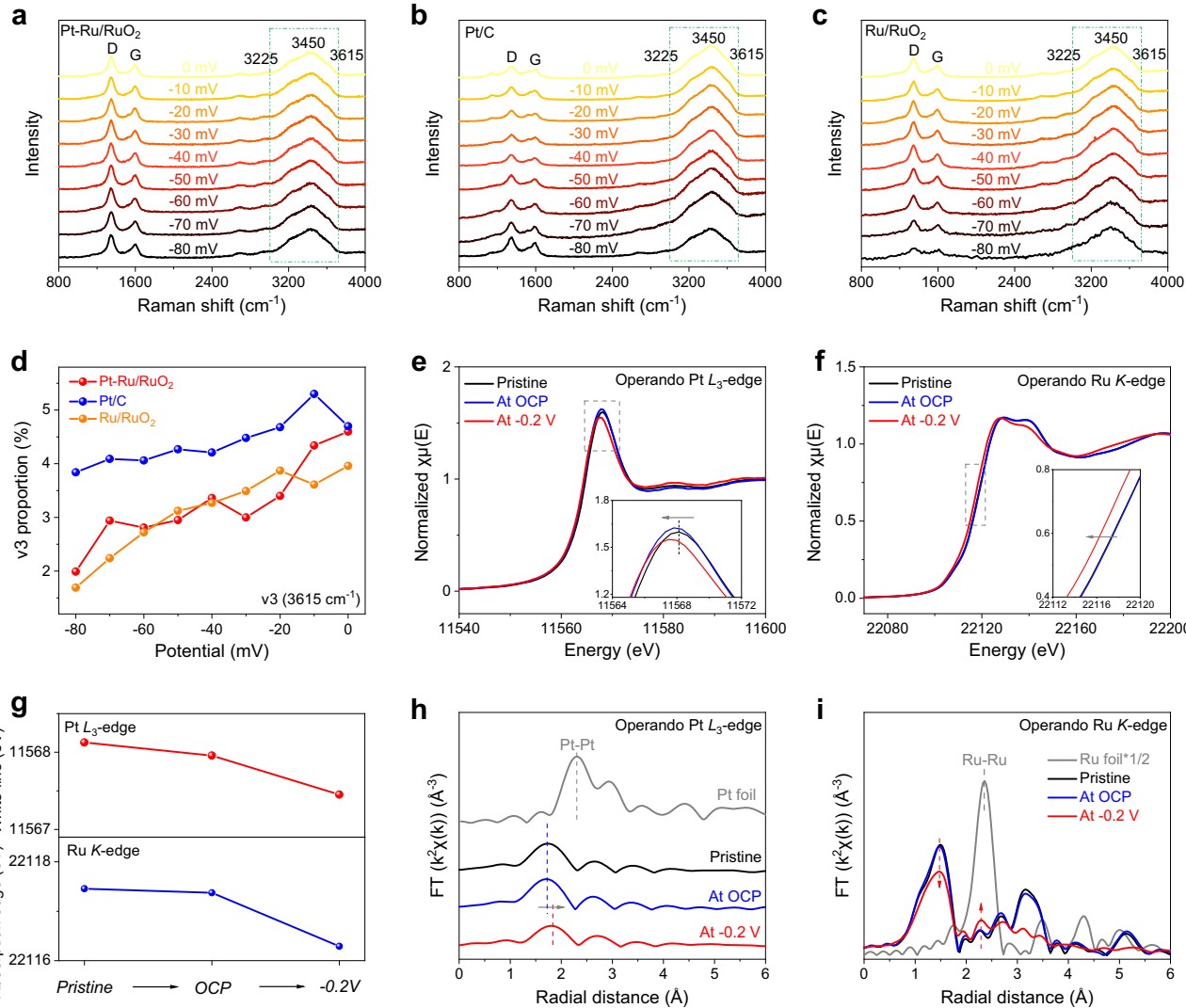

**Fig. 3 | Mechanism investigations of alkaline HER on Pt–Ru/RuO₂.** The operando Raman spectra of (**a**) Pt–Ru/RuO₂, (b) Pt/C and (c) Ru/RuO₂ under applied potentials in 1 M KOH solution. **d** The proportions of υ3 peaks at 3615 cm⁻¹ on Pt–Ru/RuO₂, Pt/C and Ru/RuO₂. **e** The Pt $L_3$-edge XANES and (**f**) the Ru $K$-edge XANES of Pt–Ru/RuO₂ measured in 1 M KOH under pristine state, OCP and HER operating condition of −0.2 V. **g** The white line peak positions of Pt $L_3$-edge XANES and the absorption edge positions of Ru $K$-edge XANES under applied potentials. **h** The Pt $L_3$-edge EXAFS and (**i**) the Ru $K$-edge EXAFS under pristine state, OCP and HER operating condition of −0.2 V.

Pt–Ru/RuO₂ and Ru/RuO₂ were constructed and energetically optimized (Supplementary Fig. 43). As depicted in Fig. 4a, the charge density difference of Pt–Ru/RuO₂ shows a significant charge redistribution along the chemical bonds due to the different electronegativity, where the electron tends to move from the surrounded Ru to central Pt atoms[57]. This result confirms an efficient electronic interaction between the isolated Pt atoms and the Ru/RuO₂[58]. It is also found an obvious electron transfer from Ru to RuO₂ on the Ru/RuO₂ heterostructure, manifesting that Ru can regulate the electronic structure of RuO₂ and results in an enhanced catalytic ability (Supplementary Fig. 44). In addition, the optimized electronic structure induced by the Pt single atoms can be confirmed by the projected density of states (PDOS), which show higher occupation near the Fermi level ($E_f$) on Pt–Ru/RuO₂ compared with the Ru/RuO₂, suggesting the accelerated electron transfer and promoted conductivity caused by Pt 5$d$ orbital contribution[45]. Moreover, the corresponding $d$-band centers are calculated from the PDOS. Specifically, the average $d$-band center values for Ru/RuO₂ and Pt–Ru/RuO₂ are −1.285 and −1.341 eV, respectively. According to the $d$-band center theory, the bonding of the reaction intermediate is weakened, and more antibonding state is

occupied with the downshifted $d$-band center position. Thus, the active sites on Pt–Ru/RuO₂ are quickly exposed, and the HER process is boosted (Fig. 4b)[56]. Subsequently, the mechanisms of different active sites in Pt–Ru/RuO₂ were investigated by employing DFT to explore their energetic properties. Since the Volmer step is the precondition and main barrier for catalyzing the HER in alkaline media, it is vital to measure the adsorption energy of H₂O on each active site, which is a key factor to evaluate the ability of accelerating water dissociation process[6]. As presented in Fig. 4c, it is clearly seen that the Ru site of RuO₂ with a higher adsorption energy (−0.594 eV) is thermodynamically more favorable for H₂O* adsorption compared with the Pt sites (−0.359 eV) and Ru sites (−0.207 eV), indicating that H₂O is more possible to be absorbed on the RuO₂[59,60]. The H₂O* dissociation free energy in Supplementary Fig. 45 further indicates that Pt single atoms are difficultly to break the H₂O molecule due to the largest energy barrier. In addition, although Ru and RuO₂ have similar energy barriers, considering the stronger ability of adsorbing H₂O on RuO₂, the water dissociation is more efficient and easier to be completed by RuO₂ rather than Ru. On the other hand, the H* adsorption free energy on all sites in Pt–Ru/RuO₂ were also calculated to examine their abilities for

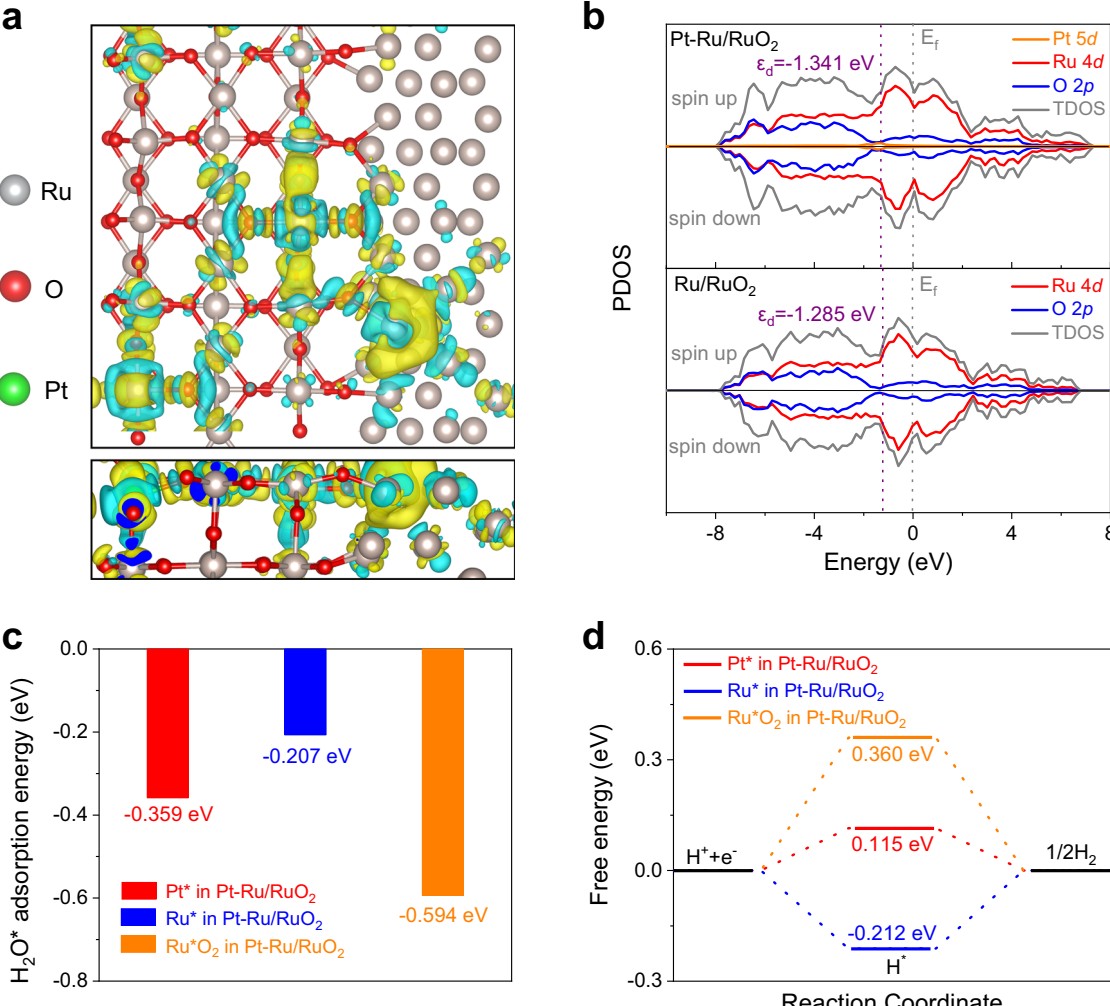

**Fig. 4 | DFT calculations of Pt–Ru/RuO₂. a** The top and side views of charge density difference on Pt–Ru/RuO₂ heterostructure model, blue and yellow areas represent the electron-accumulated region and the electron-depleted region, respectively. **b** The PDOS spectra and corresponding *d*-band centers of Pt–Ru/RuO₂ and Ru/RuO₂ models. The (**c**) H₂O* adsorption energy and (**d**) H* adsorption free energy values for Pt, Ru and RuO₂ sites in Pt–Ru/RuO₂.

activating the following H* combination step. As shown in Fig. 4d, compared to the other active sites, Pt sites deliver the most optimal value of 0.115 eV, demonstrating that the incorporated Pt single atoms are favorable for boosting the H* combination step[61–63]. It is also noted that the H* adsorption-free energy on Ru is closer to zero than RuO₂, which suggests that Ru also contributes to the H* adsorption and H₂ generation. Therefore, the finding disclosed by DFT calculations is highly consistent with the experimental results, which jointly demonstrate that the Pt single atoms, Ru and RuO₂ cooperatively enhance the alkaline HER performance of Pt–Ru/RuO₂.

## AEMWE performance of Pt–Ru/RuO₂

Encouraged by the excellent HER activity, we further constructed the membrane electrode assembly (MEA) in AEMWE by employing Pt–Ru/RuO₂ as the cathode to evaluate its performance in the practical water splitting. The mass loading of Pt–Ru/RuO₂ was controlled as 0.5 mg cm⁻², meanwhile the benchmark Pt/C (20 wt%) with 0.5 mg$_{Pt}$ cm⁻² was also tested in AEMWE for the comparison. In the preliminary tests, Ni felt was directly used as the anode (Supplementary Fig. 46). As shown in Fig. 5a, Pt–Ru/RuO₂ exhibits higher current densities than commercial Pt/C at the same cell potentials. The catalytic performances can be further enhanced by adopting the inexpensive NiFe layered double hydroxides (LDH) in the anode side. As a result, the polarization curves show that NiFe LDH +

Pt–Ru/RuO₂ only requires 1.90 V and 1.77 V to reach the current density of 1 A cm⁻² before and after iR correction, respectively, which is much lower than those on NiFe LDH + commercial Pt/C (1.98 V and 1.84 V). In addition, as depicted in Fig. 5b, the mass activities of Pt–Ru/RuO₂ reach 3.60 and 4.77 A mg⁻¹$_{Pt+Ru}$ at 2.0 V and 2.1 V, respectively, which are more than two-order of magnitude higher than those of commercial Pt/C (no iR correction). On the basis of the remarkable performances, we further consider that the cost of Ru is well below Pt, thus Pt–Ru/RuO₂ possesses a superior economic efficiency compared with the commercial Pt/C. Impressively, after normalizing the current density to the price of the noble metal, the price activities of Pt–Ru/RuO₂ are 188.2 and 247.1 A dollar⁻¹ at 2.0 V and 2.1 V, respectively, reaching almost 3 times higher than those of commercial Pt/C (72.0 and 92.0 A dollar⁻¹) (Fig. 5c). The advantages of catalytic activity and cost-saving on Pt–Ru/RuO₂ illustrate its feasibility and potential to be applied as the practical catalyst for the hydrogen production in alkaline.

## Discussion

In conclusion, a highly active and stable electrocatalyst of Pt–Ru/RuO₂ towards alkaline HER is elaborately fabricated in this work. Comprehensive experiments and operando characterizations, as well as DFT calculations, verify that all the isolated Pt atoms, Ru and RuO₂ play vital roles during the catalytic reactions. Specifically, the

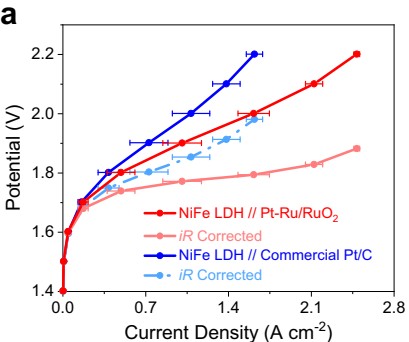

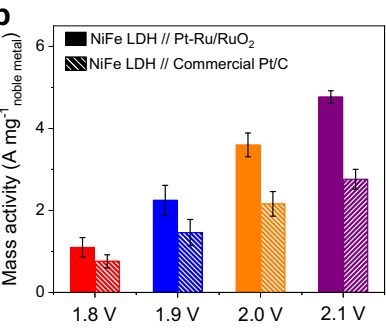

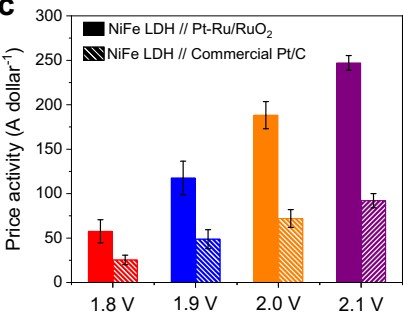

**Fig. 5 | AEMWE performances of Pt–Ru/RuO₂. a** The AEMWE polarization curves with and without iR correction on NiFe LDH // Pt–Ru/RuO₂ and NiFe LDH // commercial Pt/C. The comparisons of (**b**) mass activities and (**c**) price activities between NiFe LDH // Pt–Ru/RuO₂ and NiFe LDH // commercial Pt/C at various cell potentials (without iR correction). Note: error bars represent the standard deviation of three independent measurements.

sluggish water dissociation step is primarily accelerated by $RuO_2$ while the following hydrogen combination step is facilitated by Pt single atoms and Ru, collectively contributing to the improved HER in 1 M KOH. As a result, Pt–Ru/RuO₂ exhibits remarkable overpotentials of 18 mV and 63 mV at the current densities of 10 mA cm⁻² and 250 mA cm⁻², respectively, superior to most of the recently reported catalysts. Especially, the cost-based activity of Pt–Ru/RuO₂ can reach more than 16 times higher than those of commercial Pt/C and Ru/C, showing its good cost-effectiveness. In addition, the HER stability of Pt–Ru/RuO₂ can be prolonged to over 100 h with a negligible activity loss. Furthermore, the integration of Pt–Ru/RuO₂ and NiFe LDH in the AEMWE exhibits a superior price activity of 247.1 A dollar⁻¹ at 2.1 V, reaching almost 3 times higher than the Pt/C-based counterpart. This finding offers a prospective strategy of composite engineering to tip the balance between the high activity and economic efficiency when fabricating the noble metal-based catalysts toward practical $H_2$ production.

## Methods
### Materials
Ruthenium trichloride ($RuCl_3$, solid, 99.5%), platinum tetrachloride ($PtCl_4$, solid, 99.9%), ruthenium oxide ($RuO_2$, solid, 99%), potassium hydroxide (KOH, solid, ≥98%), and commercial Ru/C catalyst (5 wt%) were purchased from Innochem. Sodium nitrate ($NaNO_3$, solid, ≥99%) and isopropanol ($C_3H_8O$, liquid, ≥ 99.7%) were purchased from Sinopharm Chemical Reagent Co. Ltd (Shanghai, China). Commercial Pt/C catalyst (20 wt%) was obtained from Johnson Matthey (JM) corporation.

### Catalyst preparation
In a typical synthesis, 5 g $NaNO_3$ was initially heated at 400 °C for 30 min in an oven under air atmosphere, then 50 mg $RuCl_3$ powder was poured into the molten $NaNO_3$ and kept at 400 °C for 10 min. Afterwards, the cooled solid was washed by water 3 times to remove the $NaNO_3$ and dried overnight to obtain the synthesized $RuO_2$. Then 50 mg synthesized $RuO_2$ was dispersed in 18 mL water and 7 mg $PtCl_4$ dissolved in 2 mL water was dropwise added into the above solution under continuously stirring. After the formation of a homogeneous mixture, it was heated to 90 °C and kept stirring for 16 h. Finally, the Pt species absorbed $RuO_2$ was washed by water for 3 times and collected by centrifugation, followed by calcination in Ar at 450 °C for 2 h with a heating rate of 5 °C min⁻¹ to obtain the Pt–Ru/RuO₂. The Ru/RuO₂ was prepared by a similar method as Pt–Ru/RuO₂, except that the $PtCl_4$ precursor was not added.

### Characterization
The TEM and HRTEM images were determined by a JEOL 2100F instrument at a working voltage of 200 kV. AC HAADF-STEM images

and corresponding elemental mappings were collected on a Thermo Scientific Themis Z microscope operated at 200 kV. SEM-EDS spectra were collected on a Zeiss Supra 55 at an acceleration voltage of 5 kV. The mass fraction of Pt in Pt–Ru/RuO₂ and the dissolved ion concentrations after stability tests were determined by ICP-OES on an Agilent ICP-OES 730.

The XRD data was collected on a Bruker D8 Advance powder diffractometer (operating at 40 kV, 40 mA) equipped with a Cu-K$_\alpha$ source ($\lambda 1 = 1.5405$ Å, $\lambda 2 = 1.5443$ Å) and fitted with a beryllium window at room temperature. Rietveld refinements for XRD data were conducted by using the Full-Prof program, and the refined parameters included the background parameters, line shift errors (zero shift), Caglioti coefficients (U, V and W), scale factor, lattice parameters, atomic position, atomic rate occupancy and isotropic atomic displacement parameters.

The XPS was carried out on a Thermo Scientific Escalab 250Xi with an Al-Kα source. XAS of Ru $K$-edge and Pt $L_3$-edge were conducted at the 44A beamline of the National Synchrotron Radiation Research Center (NSRRC) in Taiwan. The operando XAS experiments of Ru $K$-edge and Pt $L_3$-edge were performed at the 44A beamline of the NSRRC with a custom-built operando XAS instruments. All XAS experiments were collected under the room temperature and analyzed by using the standard program Demeter.

The operando Raman spectra were collected on a HORIBA XploRA PLUS Raman spectrometer with an electrochemical cell, catalysts without carbon black were directly dropped on glassy carbon plate as the working electrode, calomel electrode and carbon rod were used as the reference electrode and counter electrode, respectively. It operated from 800 to 4000 cm⁻¹ and 0 to −80 mV vs. RHE in 1 M KOH solution to evaluate the water dissociation ability of catalysts.

### Electrochemical Measurement
The HER performance evaluation was carried out on the CHI 760E electrochemical workstation with a rotating disk electrode (RDE) configuration (Pine Research Instrumentation). A glassy carbon electrode with the surface area of 0.196 cm², calomel electrode and graphite rod were selected as the working, reference and counter electrode, respectively. Before testing, 5 mg sample (Pt–Ru/RuO₂ or Ru/RuO₂) and 5 mg carbon black (Vulcan XC-72R) were mixed in 1.9 mL isopropanol solution and sonication for 1 h, followed by adding 0.1 mL Nafion solution (5 wt%, Aldrich) in the above mixture and sonication for 5 min to produce a homogeneous catalyst ink. Then, 10 μL of the catalyst ink was dropped onto the working electrode and dried under infrared light. The mass loadings of metals for these prepared samples were controlled at 0.128 mg cm⁻². For comparison, the Pt and Ru mass loadings for commercial Pt/C (20 wt %) and commercial Ru/C (5 wt%) were also controlled at

0.128 mg cm$^{-2}$ on the working electrode as well. All HER potentials were calibrated with respect to the RHE scale (Supplementary Fig. 12). The calibration was performed by using a Pt wire as the working electrode, the calomel electrode as the reference electrode and a graphite rod as the counter electrode in a H$_2$-saturated electrolyte (1 M KOH). CV scanning was carried out with a scan rate of 1 mV s$^{-1}$ and the average potential at which the currents crossed zero was the calibration value. LSV with iR correction was conducted at a scan rate of 5 mV s$^{-1}$ under room temperature to evaluate the HER activity. The iR correction was auto-compensated by the electrochemical stations and the iR correction value was controlled as 95% for all polarization curves. Tafel slopes were determined by plotting the overpotential vs. the logarithm of current density (log | j |). The Tafel slopes were probed on Pt−Ru/RuO$_2$, Pt/C, Ru/C, Ru/RuO$_2$ and RuO$_2$ within the overpotential ranges of 0.01–0.033 V, 0.023–0.073 V, 0.031–0.086 V, 0.083–0.14 V and 0.11–0.22 V, respectively. Stability was evaluated by using catalysts loaded carbon paper (1 cm$^2$) with mass loadings of 0.128 mg cm$^{-2}$ to carry out chronopotentiometry at the constant HER current densities of 10 mA cm$^{-2}$ and 250 mA cm$^{-2}$, respectively. Cyclic stability tests were carried out by repeating 5000 CV cycles on the studied samples. EIS was measured on the Bio-Logic SP-200 workstation in the frequency range from 0.01 to 100 kHz at −50 mV vs. RHE; the operando EIS was measured on all samples in the frequency range from 0.01 to 100 kHz at various applied potentials. Corresponding equivalent circuits were established and fitting resistance values were obtained through the EC-lab Software Analysis.

## Calculation of the electrochemically surface area
ECSAs were calculated by evaluating the electrochemical double-layer capacitance (C$_{dl}$) on catalytic surface of studied catalysts based on Eq. (1):

$$ECSA = R_f S = \frac{C_{dl}}{C_s} S \quad (1)$$

where C$_{dl}$ was measured from the scan-rate-dependent CVs in the non-Faradaic region from 0.5 to 0.6 V vs. RHE in 1 M KOH with scan rates of 20, 40, 60, 80 and 100 mV s$^{-1}$. S represents the real surface area of the smooth metal electrode, which was generally equal to the geometric area of the glassy carbon electrode (S = 0.196 cm$^2$). The specific capacitance (C$_s$) for a flat surface was generally considered to be in the range of 20–60 μF cm$^{-2}$. In this work, a C$_s$ value of 60 μF cm$^{-2}$ was accepted. ECSAs were also measured by H$_{upd}$ method, in which all catalysts were scanned by CV in N$_2$-saturated 1 M KOH solution with a scan rate of 50 mV s$^{-1}$. CO stripping tests were conducted by bubbling the CO gas directly on the catalyst-coated electrode for 20 minutes in 1 M KOH solution, followed by placing the electrode in the fresh KOH electrolyte and scanning CV to remove the absorbed CO. The scanning rate was controlled as 50 mV s$^{-1}$.

## Calculation of HER Faradaic efficiency
Pt−Ru/RuO$_2$ was dropped on carbon paper to conduct HER in a sealed cell. The applied potentials on the sample to generate H$_2$ were −0.01 V, −0.05 V, −0.1 V, −0.2 V and −0.3 V vs. RHE, respectively. The amounts of generated H$_2$ were recorded by Agilent 7890B equipped with a 5 Å molecular sieve column thermal conductivity detector and Ar was used as the carrier gas. H$_2$ amount was determined by plotting the generated H$_2$ peak area into the established standard fitting curves. The HER FE of Pt−Ru/RuO$_2$ was

calculated based on Eq. (2):

$$FE = \frac{neF}{Q} \quad (2)$$

where e is the number of electrons transferred for generating H$_2$, Q is the total charge, n is the amount of generated H$_2$ (in moles) and F is the Faradaic constant.

## Computational details
All the structural optimizations and adsorption energy calculations were carried out based on the density-functional theory (DFT) as implemented in Vienna Ab initio Simulation Package (VASP)[64]. The projector-augmented wave (PAW) method was implemented to calculate the interactions between the ionic cores and valence electrons[65]. Perdew−Burke−Ernzerhof (PBE) approach of spin-polarized generalized gradient approximation (GGA) was used to describe the exchange-correlation energy[66]. All DFT calculations were performed for the (1 0 −1 1)-Ru/(1 1 0)-RuO$_2$ heterostructure model. The atomistic structures of (1 0 −1 1)-Ru and (1 1 0)-RuO$_2$ were generated from Materials Project (MP) database[67]. Besides, 3 Ru atoms for Ru/RuO$_2$ heterostructure model were randomly replaced by Pt atoms with the lowest-energy configuration to approximately simulating 1.0 % atomic content and construct the Pt−Ru/RuO$_2$ heterostructure model. The cutoff plane-wave kinetic energy of 520 eV was used in calculations with the 1×1×1 k-point grid size. The conjugate-gradient method was used in electronic relaxation, and all DFT convergence parameters were set consistent with the Materials Project (MP)[63]. The adsorption energy (E$_{ads}$) is defined as

$$E_{ads} = E_{ads/slab}^{tot} - E_{slab}^{tot} - E_{ads}^{tot} \quad (3)$$

where E$_{slab}^{tot}$ is the energy of slab, E$_{ads}^{tot}$ is the energy of free adsorbate, and E$_{ads/slab}^{tot}$ is the total energy of adsorbate-slab system in the equilibrium state.

The free energy (ΔG) of adsorbed H* and H$_2$O* is defined as

$$\Delta G = \Delta E + \Delta E_{ZPE} - T\Delta S \quad (4)$$

where ΔE is the adsorption energy of slab, ΔE$_{ZPE}$ is the difference of zero-point energy between adsorbed state and the gas phase, TΔS is the entropy contribution to the reaction at T = 298.15 K.

## MEA preparation for AEMWE single-cell measurements
(AEMION$^{TM}$ Membrane, AEMION$^{TM}$ Ionomer) The membrane electrode assembly (MEA) was prepared by manual spray coating technique of catalyst-coated membranes (CCM) as described by Klingenhof et al.[68]. For spray coating the membrane (AEMION$^{TM}$) was fixed on a commercial heating vacuum plate equipped with temperature control (Carbon and Fuel Cell) and a compressor (Welch). The table temperature was set to 55 °C. Furthermore, a mask (5 cm$^2$) was used to ensure vis-a-vis coating of the anode and cathode. The applied anode and cathode ink consisted in both cases of 50 mg catalyst powder, 50 μL ultrapure water, 3 mL i-PrOH and 460 mg ionomer solution (AEMION$^{TM}$, 5 wt% solution in EtOH and i-PrOH). In the case of the anode, an overall amount of 500 μL ink were sprayed on the membrane, aiming at a NiFe-LDH loading of 1 mg cm$^{-2}$. At the cathode catalyst ink with the loading amount of 0.5 mg cm$^{-2}$ were spray coated on the membrane. After spray coating, the as-prepared CCM was dried for ten minutes at 55 °C followed by cooling to RT and weighting. To achieve a satisfying loading spraying and weighing step were carried out several times with low amounts of the catalyst ink. The anode was always spray

coated first. To adjust a suitable loading and provide suitable comparisons between commercial Pt/C reference catalyst and the manufactured catalyst measurements without NiFe-LDH as anode catalyst were conducted avoiding a more complex system. After proper adjustment of the cathode catalyst loading, measurements with additional NiFe-LDH catalyst were conducted. The measurements were carried out using a Greenlight test station with a 5 cm$^2$ AEMWE single-cell setup. The setup consists of the prepared MEA, furthermore oxygen and hydrogen side are equipped with PTFE gaskets, porous transport layers (PTL, Nickel Felt at the Anode and Carbon Cloth at the cathode), endplates with electrolyte ports, bi-potential plates (parallel flow fields, carbon plate for the hydrogen side and titanium plate for the oxygen side). The CCMs were pre-treated in 2 M KOH over 2 h before the measurement. To achieve the best performance, the spray-coated CCMs were anion-exchanged in 2 M KOH three times, each 20 min. To remove the remaining surplus OH- the anion-exchanged membrane was washed with DI water three times for 20 min. Before evaluating the activity by applying polarization curves, the CCM was activated by potential cycling (50 mV between 1.1 V and 1.7 V cell voltage) and a potential holding step at 1.83 V over 1 h. The single-cell measurements were conducted at 60 °C in 0.1 M KOH solution.

## Data availability

All relevant data are available from the corresponding authors on request. Source data are provided with this paper.

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

## Acknowledgements

The work leading to these results has received funding from the National Natural Science Foundation of China (22179098) and the Fundamental Research Funds for the Central Universities to J.M. The project leading to this application has received funding by European Union's HORIZON.3.1 - The European Innovation Council (EIC) Programme through the grant agreement 101071111 - ANEMEL. P.S. and M.K. acknowledge the financial support by the federal ministry for education, research and development (Bundesministerium für Bildung und Forschung, BMBF) under Grant Numbers 03SF0613D and 03HY130B in the collaborative research projects "AEMready" and "AEMDirekt". Z.H. acknowledges the support from the Max Planck-POSTECH-Hsinchu Center for Complex Phase Materials. M.Y. acknowledges the support from the National Natural Science Foundation of China (52302302).

## Author contributions

J.M. conceived and supervised the research. Y.Z. and J.M. designed the experiments. Y.Z., M.K., T.K., G.W., Y.P., S.L., J.H., J.L., Z.H., P.S. and J.M. performed the experiments and data analysis. Y.Z., C.G., L.S. and M.Y. performed the theoretical computations and analyzed the theoretical results. Y.Z., H.J., L.X., W.H., C.P., Z.H., P.S. and J.M. participated in various aspects of the experiments and discussions. Y.Z., Z.H., P.S. and J.M. wrote the paper. All authors discussed the results and commented on the manuscript.

## Funding

## Competing interests

The authors declare no competing interests.
