## [Peer Review File · Nature Communications]

Facilitating alkaline hydrogen evolution reaction on the hetero-interfaced Ru/RuO₂ through Pt single atoms dopingREVIEWER COMMENTS

Reviewer #1 (Remarks to the Author):

In this manuscript, the authors designed Pt single atom supported on heterointerfaced Ru/RuO₂ support for alkaline HER. The Pt-Ru/RuO₂ catalyst exhibits outstanding mass activity compared commercial Pt/C. The reaction mechanism is further unveiled by operando spectroscopic analysis, which is pretty interesting. However, there is still some scientific issues should be clarified.

1. The Pt⁰ is derived from the Pt single atoms dispersed on the Ru. However, DFT calculations shows the electron transfer from Ru to Pt, which will lead a positive oxidation state of Pt single atoms. These two conclusions are contradictory. I think the Pt⁰ is maybe originated from few Pt clusters or nanoparticles during the reduction of Pt precursor.
2. The Tafel slope of Pt-Ru/RuO₂ is an extremely low value of 18.5 mV dec⁻¹, which is even lower than the theoretical value of the Tafel step. This result suggests that the reaction step is not traditional Tafel or Heyrovsky step. However, the authors do not explain this phenomenon.
3. The operando Raman shows two peaks of D and G band at 1400 cm⁻² and 1600cm⁻², which is attributed to the carbon signal. Therefore, I doubt that the lasers focus on carbon black instead of the surface of catalyst.
4. According to the DFT results, the H₂O adsorption energy of Ru* is higher than that of Pt*, indicating that Pt sites dispersed on metallic Ru are favorable for the water dissociation instead of surrounding Ru sites. Therefore, only RuO₂ play the role of water dissociation. The hypothesis that H₂O is more possible to be absorbed and cleaved by the Ru/RuO₂ supports is actually wrong.

Reviewer #2 (Remarks to the Author):

In this manuscript, the authors demonstrate that the platinum single atoms (SAs) doped Ru/RuO₂ support (Pt-Ru/RuO₂) catalyst exhibits a high HER activity and an excellent stability.

Despite many physical/chemical characterizations and theoretical calculations were made in this work, we are afraid that several inconsistencies in mechanism and lack of novelty prevent us from a favorable recommendation of this study for publication. Some specific comments are listed as follows.

1. Enhancing the activity and stability of PGM and its oxides for alkaline HER by incorporation of single metal atoms is a common strategy. According to my knowledge, platinum single atoms (SAs) doped Ru/RuO₂ support have been investigated in many electrochemical reactions including methanol oxidation, oxygen reduction and hydrogen oxidation reactions. (Nature Communications volume 12, Article number: 5235 (2021); Croat. Chem. Acta 2017, 90(2), 225–230). Therefore, the novelty of this manuscript is not high enough.
2. Chronopotentiometric measurements of Pt-Ru/RuO₂ and Pt/C at current densities of 10 mA cm⁻² and 250 mA cm⁻² were measured on carbon paper. How about the stability of catalysts is without carbon paper?
3. The author claimed that the atomic ratio of Pt is 1.36% in Pt-Ru/RuO₂ and Pt is the active sites for HER instead of Ru. How could such limited Pt active sites contribute to the high current density under negative potential? If this statement is correct, increasing or decreasing the amount of Pt single atoms should also have a linear relationship with their current density. However, no relevant control experiments were seen in the article.
4. The author claimed the significant decline of activity on Pt-Ru/RuO₂ after the addition of SCN⁻, implying the essential role of Pt single atom toward activating HER. However, this experiment cannot rule out the possibility of Ru as an active site, as Ru can also be poisoned by SCN⁻. Besides, we also noticed that Ru/RuO₂ samples themselves exhibit high HER activity (Figure 2a). Therefore, the

assumption of active sites and mechanisms in this article is not accurate.

5. There are some DFT calculations in this study, however, the author only calculated H* and H₂O* adsorption energy. The free energy diagram for whole HER should be provided.

6. In the operando Raman spectra, the author claimed they observe that the ν_3 peak proportions on both Pt-Ru/RuO₂ and Ru/RuO₂ display a dramatically faster downward trend than that of Pt/C with the increased HER potentials. However, in Figure 3a to 3c, we could not see this trend, the ν_3 peak is very weak in all of the samples.

7. In the operando XANES results, the author only measured one negative potential (-0.2V). In fact, under such a negative overpotential, most metal oxides will be reduced, so this result cannot reflect the actual changes in active sites. The catalyst reported by the author exhibits high activity even at a very low negative potential (-0.018V), therefore the author should test in-situ XANES near this potential in stages, rather than just -0.2V.

Reply to Reviewers on a point-by-point basis

Reviewer #1 (Remarks to the Author):

In this manuscript, the authors designed Pt single atom supported on heterointerfaced Ru/RuO₂ support for alkaline HER. The Pt-Ru/RuO₂ catalyst exhibits outstanding mass activity compared commercial Pt/C. The reaction mechanism is further unveiled by operando spectroscopic analysis, which is pretty interesting. However, there is still some scientific issues should be clarified.

Response: We thank the reviewer for the very positive statements: “*The reaction mechanism is further unveiled by operando spectroscopies analysis, which is pretty interesting.*”, and many valuable suggestions to improve the quality of our manuscript.

1. The Pt⁰ is derived from the Pt single atoms dispersed on the Ru. However, DFT calculations shows the electron transfer from Ru to Pt, which will lead a positive oxidation state of Pt single atoms. These two conclusions are contradictory. I think the Pt⁰ is maybe originated from few Pt clusters or nanoparticles during the reduction of Pt precursor.

Response: We thank the reviewer for carefully reading our manuscript and this insightful comment. As to the question of the formation of Pt clusters, we have no experimental evidence for the formation of Pt clusters. If Pt clusters or nanoparticles were generated, our bulk-sensitive element-sensitive EXAFS analyses should reveal obvious Pt-Pt coordination shells, which is absent in Figure 1h [*J. Am. Chem. Soc.* **144**, 15529-15538 (2022); *Nat. Commun.* **14**, 3767 (2023); *J. Am. Chem. Soc.* **145**, 21432-21441 (2023)]. Furthermore, we note that we did not observe any aggregated Pt in the HAADF-STEM images, which is a quite local method. All these make us confident that Pt exists indeed as single atoms dispersed in Pt-Ru/RuO₂.

The detailed atomic states of Pt⁰ and Ru atoms are certainly very complex. Based on our analyses, we tend to believe at this point that the single-atom alloy is formed when Pt atoms are homogeneously dispersed in regions of metallic Ru. Based on prior reports in [*Nat. Nanotechnol.* **18**, 611-616 (2023); *Chem. Soc. Rev.* **50**, 569-588 (2021); *Nat. Mater.* **17**, 1033-1039 (2018)], where this process has been investigated in more details: the single-atom alloy shows similar properties as the pure metal with free-atom states and electron transfer between different atoms through the metallic bond. Thus, Pt single atoms in the single-atom alloy feature the metallic characteristics and have a valence state of zero.

Change: We have added an explicit and detailed clarification from Line 150 to Line 153 in the revised manuscript: “The single-atom alloy is formed when Pt single atoms are homogeneously dispersed in the region of Ru, in which Pt features the metallic characteristics and thus has a valence state of zero.³⁶⁻³⁸ Meanwhile the oxidized Pt (Supplementary Fig. 8d) is primarily originated from the Pt single atoms coordinated by O atoms on the RuO₂.”.

2. The Tafel slope of Pt-Ru/RuO₂ is an extremely low value of 18.5 mV dec⁻¹, which is even lower than the theoretical value of the Tafel step. This result suggests that the reaction step is not traditional Tafel or Heyrovsky step. However, the authors do not explain this phenomenon.

Response: This is a valuable comment. In electrocatalysis, Tafel slope can provide insights for analyzing the elementary steps and reaction mechanisms. In fact, the canonical theoretical Tafel values (120, 40, 30 mV dec⁻¹) are calculated based on ideal microkinetics and many assumptions are adopted, which do not apply as such in most of the real systems. For instance, the surface and bulk concentrations of H⁺ are assumed to be the same in the equation of deducing the theoretical Tafel values, however, it's indeed different in the realistic electrochemical conditions due to the binding capacity of the electrocatalysts, which will cause the extra H₂ current and lower the Tafel slope. In addition, the Tafel behavior of real electrocatalytic interfaces is also influenced by intermediates coverages and other factors such as solvent and electrolyte ion interaction. [*Int. J. Hydrogen Energy* **44**, 19484-19518 (2019); *Sci. Rep.* **5**, 13801 (2015)]. Therefore, the derived Tafel slope values in practical systems are not exactly the same as the theoretical values.

In our work, it can be seen that the Tafel slope is reduced from 58 mV dec⁻¹ on Ru/RuO₂ to 18.5 mV dec⁻¹ on Pt-Ru/RuO₂ after the introduction of Pt single atoms with high hydrogen coverage and binding ability. It is reported that the largely enhanced coverage of H_{ads} intermediates usually enhances the reaction kinetics, causing the decreased Tafel slope value and driving the catalysts to follow the Volmer-Tafel mechanism in alkaline HER [*ACS Catal.* **13**, 4752-4759 (2023); *Nat. Commun.* **14**, 5363 (2023)]. Thus, we can conclude that Pt-Ru/RuO₂ follows the Volmer-Tafel mechanism in alkaline HER. In addition, similar low Tafel slopes have also been reported such as 15 mV dec⁻¹ in *Adv. Mater.* **34**, 2206368 (2022); 21.7 mV dec⁻¹ in *Adv. Energy Mater.* **12**, 2200029 (2022); 22 mV dec⁻¹ in *Nat. Commun.* **13**, 5497 (2022); 21.6 mV dec⁻¹ in *Adv. Mater.* **35**, 2207114 (2023) and 18.6 mV dec⁻¹ in *Energy Environ. Sci.* **16**, 574-583 (2023), in which the authors attributed these low values to the Volmer-Tafel mechanism of electrocatalysts in alkaline HER.

Change: We have added the clarification from Line 210 to Line 212 and corresponding references in the revised manuscript: “In addition, such a low Tafel slope value on Pt-Ru/RuO₂ also indicates that it follows the Volmer-Tafel mechanism in alkaline HER after the incorporation of Pt single atoms with high hydrogen coverage and binding ability.^{43,44}”.

3. The operando Raman shows two peaks of D and G band at 1400 cm⁻¹ and 1600 cm⁻¹, which is attributed to the carbon signal. Therefore, I doubt that the lasers focus on carbon black instead of the surface of catalyst.

Response: We thank the reviewer for noticing this point. In the process of testing the operando Raman, no carbon black was involved. The catalysts were directly cast onto the glassy carbon plate used as the working electrode. Thus the two peaks of D and G band at 1400 cm⁻¹ and 1600 cm⁻¹ derive from the glassy carbon substrate, not from carbon black. To experimentally confirm this, as shown in Supplementary Figure 32a, we add the ex-situ Raman test directly on the glassy carbon plate without electrocatalyst and electrolyte, the D and G band signals can be clearly found. Since Raman spectra are detected by transmitting laser through the samples to obtain signal, the D and G band signals from the glassy carbon plate are unavoidable.

Furthermore, in order to verify that Raman lasers were focus on the surface of electrocatalysts, the operando Raman tests under various potentials in 1 M KOH solution are carried out on the glassy carbon plate without electrocatalyst. As depicted in Supplementary Figure 32b, due to its inert

property toward electrocatalysis, almost no interfacial water peaks can be observed on the glassy carbon plate at around 3500 cm^{-1} under operating conditions. In sharp contrast, these signals evidently appear in the operando Raman spectra of electrocatalysts (Figure 3a). Based on above experiments, we conclude that the D and G bands are attributed to the glassy carbon plate and the Raman lasers are evidently focus on the catalyst surface to monitor the dynamic evolution of interfacial water.

Change: We have added the Raman spectra of glassy carbon plate under ex-situ condition and various applied potentials in Supplementary Figure 32. Corresponding descriptions have also been included from Line 331 to Line 337 in the revised manuscript: “The two carbon signals of D and G bands at around 1400 and 1600 cm^{-1} are derived from the glassy carbon plate (Supplementary Fig. 32a). In addition, in order to confirm that the Raman laser was focus on the surface of electrocatalyst, the operando Raman tests under various potentials in 1 M KOH solution were conducted on the glassy carbon plate without electrocatalyst. It can be clearly seen that almost no interfacial water signals (3500 cm^{-1}) can be found in Supplementary Fig. 32b, which is due to the inert property of the glassy carbon plate towards the catalytic reaction.” Corresponding test details are supplemented in the Methods section.

Supplementary Figure 32. (a) The ex-situ Raman spectrum of the glassy carbon plate without catalyst and electrolyte. (b) The operando Raman spectrum of the glassy carbon plate in the electrolytes under various applied potentials.

4. According to the DFT results, the H₂O adsorption energy of Ru* is higher than that of Pt*, indicating that Pt sites dispersed on metallic Ru are favorable for the water dissociation instead of surrounding Ru sites. Therefore, only RuO₂ play the role of water dissociation. The hypothesis that H₂O is more possible to be absorbed and cleaved by the Ru/RuO₂ supports is actually wrong.

Response: We fully agree with the reviewer. As shown in Figure 4c, compared with RuO₂, both Pt and Ru show smaller H₂O* adsorption energy, thus only RuO₂ in Pt-Ru/RuO₂ should be considered as the active sites for efficiently adsorbing and dissociating the H₂O molecule. We are sorry to ignore the function of metallic Ru and suppose that the overall Ru/RuO₂ support is responsible for water dissociation. According to Tafel slope analysis, the HER mechanism is changed from Volmer step on RuO₂ (120.5 mV dec^{-1}) to Volmer-Heyrovsky step on Ru/RuO₂ (55.0 mV dec^{-1}), suggesting the metallic Ru is also responsible for accelerating the H* combination

step. This is verified by experiment results (H_{upd} and operando EIS tests), which disclose that in addition to Pt single atoms, the metallic Ru also contributes to the H^* adsorption to promote the HER activity. H^* adsorption free energy calculations in Figure 4d further indicates that Ru has a more optimized H^* binding energy than RuO_2 . Based on above investigations, we should draw the conclusion that all Pt single atoms, Ru and RuO_2 participate in the electrochemical reaction. Specifically, the water dissociation step is facilitated by RuO_2 in Pt-Ru/ RuO_2 , while both Pt single atoms and Ru boost the subsequent H^* combination step, collectively contributing to enhanced alkaline HER performances.

Change: We have corrected the related discussions in the revised manuscript.

Reviewer #2 (Remarks to the Author):

In this manuscript, the authors demonstrate that the platinum single atoms (SAs) doped Ru/ RuO_2 support (Pt-Ru/ RuO_2) catalyst exhibits a high HER activity and an excellent stability. Despite many physical/chemical characterizations and theoretical calculations were made in this work, we are afraid that several inconsistencies in mechanism and lack of novelty prevent us from a favorable recommendation of this study for publication.

Response: We thank the reviewer for carefully reading our manuscript and we appreciate these valuable comments, which help us improve the quality of the manuscript to a large extent.

Some specific comments are listed as follows:

1. Enhancing the activity and stability of PGM and its oxides for alkaline HER by incorporation of single metal atoms is a common strategy. According to my knowledge, platinum single atoms (SAs) doped Ru/ RuO_2 support have been investigated in many electrochemical reactions including methanol oxidation, oxygen reduction and hydrogen oxidation reactions. (Nature Communications volume 12, Article number: 5235 (2021); Croat. Chem. Acta 2017, 90 (2), 225–230). Therefore, the novelty of this manuscript is not high enough.

Response: We thank the reviewer for raising this concern about novelty for our presented work. And we thank the reviewer for pointing out these new references, which we will include in our study. Upon reviewing these works, we notice that our present work does provide significant novel insights and discussion with respect to the HER catalytic process, which is studied on Pt single metal atoms. In particular, the level of characterization and understanding of the mechanism of this catalyst under operating in-situ conditions is novel and has not been reported before.

We here want to highlight the novelties our study provides for the hydrogen evolution reaction process: Due to the optimized H^* binding ability, Pt is considered as the most suitable catalyst toward the HER. However, in the alkaline medium, the catalytic activity of Pt is largely limited because of its poor water dissociation ability. In addition, RuO_2 , as one of the cheapest noble metal derivatives, can effectively break the water molecule but shows insufficient hydrogen recombination ability [*Nat. Commun.* **13**, 6486 (2022)]. Based on this, we experimentally demonstrated that the alkaline HER activity and stability can be greatly enhanced after forming

the composite (Pt-Ru/RuO₂) by integrating Pt and Ru/RuO₂. In the synthesis process, Pt was designed as the single atoms to largely reduce the cost and maximum the atomic utilization, meanwhile a little fraction of metallic Ru was simultaneously generated to regulate the electronic structures of RuO₂ and contribute to the HER. In fact, after carefully reviewing the literatures, our synthesized Pt-Ru/RuO₂ with single-atom doping and heterointerfaced structure for alkaline HER has not yet been reported so far. The catalysts in literatures listed by the reviewer are Pt single atoms doped RuO₂, and their applications are MOR and ORR, respectively [*Nat. Commun.* **12**, 5235 (2021); *Croat. Chem. Acta.* **90**, 225-230 (2017)].

In addition, our synthesized Pt-Ru/RuO₂ only delivers a low overpotential of 18 mV at 10 mA cm⁻², which outperforms many recently reported Pt-based catalysts in alkaline HER [*Nat. Commun.* **14**, 1711 (2023); *J. Am. Chem. Soc.* **145**, 21432 (2023); *Energy Environ. Sci.* **16**, 4093-4104 (2023)]. Furthermore, the price activity of Pt-Ru/RuO₂ reaches almost 15 times higher than commercial Pt/C and Ru/C, which exhibits a good comprise with respect to the HER performance and cost. More importantly, compared with the half-cell electrochemical test, the anion exchange membrane water electrolyzer (AEMWE) test under a much larger operating current density is more crucial and meaningful, which can provide a more accurate evaluation of the catalyst efficiency in the practical application [*Energychem* **4**, 100087 (2022); *Carbon Neutralization.* **1**, 26-48 (2022)]. However, until now in most of literatures reporting on alkaline HER catalysts, corresponding AEMWE performances have not been investigated [*Adv. Mater.* **35**, 2301133 (2023); *Angew. Chem. Int. Ed.* **62**, e202311937 (2023); *Energy Environ. Sci.* **16**, 5220-5230 (2023); *Nat. Commun.* **14**, 2306 (2023)]. In our work, Pt-Ru/RuO₂ requires lower voltages to reach 1 A cm⁻² and exhibits a higher price activity than the benchmark Pt/C, its high activity and economic efficiency in the realistic conditions further indicate a great potential toward the practical H₂ production.

The present study also provides a new and previously unavailable characterization and understanding of the dynamic evolutions of the Pt single-atom catalysts during HER. Molecular-level insights into interfacial catalytic processes and dynamic changes of electrocatalysts offer guidelines for improved design of electrocatalysts. The novelties this study provides in terms of characterization of the material and reaction processes are as follows:

Operando Raman Spectroscopy is a highly sensitive technique to detect the existence states of adsorbed species during the reaction, which was employed in our work to evaluate the capacity of catalyst for promoting the water dissociation step. In addition, the **element-sensitive XAS experiment under operando electrochemical conditions** is the most powerful tool to monitor the real-time changes of electronic structures on catalysts for deciphering the mechanism, which was further conducted in our presented work to detect the dynamic evolutions of valence states on Pt and Ru elements under the operating potential and verify the active sites toward alkaline HER. Advanced **operando XAS** characterization has been absent in most prior work on Pt single-atom catalysts, for instance in the two works listed by reviewer.

2. Chronopotentiometric measurements of Pt-Ru/RuO₂ and Pt/C at current densities of 10 mA cm⁻² and 250 mA cm⁻² were measured on carbon paper. How about the stability of catalysts is without carbon paper?

Response: We thank the reviewer for this insightful advice. As suggested, chronopotentiometric (CP) tests of Pt-Ru/RuO₂ and Pt/C with the same loading amount (0.128 mg cm⁻²) are also tested on glassy carbon (GC) electrode (surface area: 0.196 cm²) and nickel foam (NF) (surface area: 1 cm²) for sake comparison.

Due to the flat and small surface area of the GC electrode, the generated H₂ bubbles are difficult to be released from the catalyst surface, hindering the electrochemical interaction with electrolyte [Angew. Chem. Int. Ed. **61**, e202103824 (2022); Carbon **186**, 282-302 (2022)]. Thus we conducted the CP tests at the current density of 10 mA cm⁻² for 42 hours on the GC electrode with a rotating rate of 1600 rpm (to remove the generated bubble). As shown in Figure R1, the CP test of Pt-Ru/RuO₂ is stably operated for 42 h with only 4 mV overpotential increase at 10 mA cm⁻², whereas Pt/C shows a large activity loss (85 mV) after operating for 42 h. In addition, compared with the dramatic increase of the overpotential on NF-supported Pt/C, a much smaller increase of overpotential is found on the NF-supported Pt-Ru/RuO₂ at the large current density of 250 mA cm⁻² for 150 h. Above measurements indicate that the Pt-Ru/RuO₂ still possesses an excellent HER stability in the alkaline electrolyte on both glassy carbon electrode and nickel foam, outperforming the commercial Pt/C.

Figure R1. The stability tests of Pt-Ru/RuO₂ and Pt/C (a) on glassy carbon electrode at 10 mA cm⁻² for 42 h and (b) on nickel foam at 250 mA cm⁻² for 150 h, respectively.

3. The author claimed that the atomic ratio of Pt is 1.36% in Pt-Ru/RuO₂ and Pt is the active sites for HER instead of Ru. How could such limited Pt active sites contribute to the high current density under negative potential? If this statement is correct, increasing or decreasing the amount of Pt single atoms should also have a linear relationship with their current density. However, no relevant control experiments were seen in the article.

Response: We thank the reviewer for raising this point. As suggested by the reviewer, in addition to our reported Pt-Ru/RuO₂, we have also prepared two other catalysts by halving and doubling the inputs of Pt precursor and their atomic ratios of Pt are determined to be 0.92% and 3.06% by ICP, respectively. Corresponding LSV results show that after only doping 0.92% Pt, the HER activity of Pt-Ru/RuO₂ is obviously improved compared with the Ru/RuO₂ owing to the introduction of highly active Pt sites. However, with further increasing the doped Pt amount from 1.36% to 3.06%, its activity decreases, which may be attributed to the formation of aggregated Pt

clusters [*J. Am. Chem. Soc.* **145**, 17577-17587 (2023); *Nat. Commun.* **13**, 6875 (2022)]. In fact, the relationship between the doped amount of single atoms and the catalytic performances in electrocatalysis is more likely to be volcanic rather than linear. This is because that the activity is closely related to the number of active sites involved in the electrochemical reaction, which does not indefinitely increase with the increased amount of doped single atoms. When the amount of input precursor is increased to a certain level, clusters or nanoparticles are formed, which inevitably limits the sufficient atom utilization as well as fails to induce the efficient interactions, causing the decreased catalytic performance [*Angew. Chem. Int. Ed.* **61**, e202209486 (2022)]. Therefore, the optimal Pt doped amount in Pt-Ru/RuO₂ is 1.36%, contributing to the best HER activity in 1 M KOH solution.

Actually, after doping only 1.36 at% Pt, the Pt-Ru/RuO₂ shows a significant activity increase compared with the Ru/RuO₂ (93 mV of overpotential decrease at 10 mA cm⁻²). Such an enhancement is achieved by doping Pt with optimized H* binding energy to activate HER, which is confirmed by both experiment and DFT calculations. Although the doping amount is only 1.36%, Pt species in our work are designed as the single atoms to ensure the maximum atomic utilization and efficient interaction in catalytic reaction, which are the advantages of the single-atom materials compared with conventional nanoparticle catalysts. Indeed, the phenomena of an obvious activity enhancement after single-atom doping are commonly reported in many literatures. For example, Zeng et al. decreased the HER overpotential by 263 mV after doping 1.27% Rh single atoms into NiFe LDH [*J. Am. Chem. Soc.* **145**, 17577-17587 (2023)]; Wei et al. reported that the HER overpotential of Mn₃O₄ exhibited a large decrease of 188 mV after the incorporation of only 1.58% Pt single atoms [*Energy Environ. Sci.* **15**, 4592-4600 (2022)]; Zhang et al. realized a sharp overpotential decrease of 150 mV after introducing 1.2% Pt single atoms into MXene [*Nat. Catal.* **1**, 985-992 (2018)].

Change: The LSV curves of Pt-Ru/RuO₂ with various amounts of Pt dopant have been added in Supplementary Figure 20. Corresponding discussion has also been added from Line 239 to Line 247 in the revised manuscript: “Moreover, the two other Pt-Ru/RuO₂ catalysts with various amounts of Pt dopant were prepared by halving and doubling the inputs of Pt precursor, their Pt atomic ratios are determined to be 0.92% and 3.06% by ICP, respectively. LSV results in Supplementary Fig. 20 show that after doping 0.92% Pt, the HER activity of Pt-Ru/RuO₂ is obviously enhanced compared to the Ru/RuO₂ due to the incorporation of highly active Pt single sites. However, the HER activity of Pt-Ru/RuO₂ decreases with further increasing the amount of Pt dopant from 1.36% to 3.06%, which may be attributed to the formation of aggregated Pt clusters, suggesting that the optimal amount of Pt dopant in Pt-Ru/RuO₂ toward alkaline HER is 1.36%.”

Supplementary Figure 20. The polarization curves of Pt-Ru/RuO₂ with different Pt doping amount in 1 M KOH solution. Note: error bars represent the standard deviation of three independent measurements.

4. The author claimed the significant decline of activity on Pt-Ru/RuO₂ after the addition of SCN⁻, implying the essential role of Pt single atom toward activating HER. However, this experiment cannot rule out the possibility of Ru as an active site, as Ru can also be poisoned by SCN⁻. Besides, we also noticed that Ru/RuO₂ samples themselves exhibit high HER activity (Figure 2a). Therefore, the assumption of active sites and mechanisms in this article is not accurate.

Response: After carefully reviewing the experiment, we agree with the reviewer’s comment and we are sorry about the neglect of Ru active sites toward HER. The SCN⁻ poisoning experiments on both Ru/RuO₂ and RuO₂ are added in the Supplementary Figure 28. As a result, the activity decline can be found on Ru/RuO₂ while almost no change happens on RuO₂ after being poisoned, indicating that Ru also contributes to HER. According to the Tafel slope analysis, the HER mechanism is changed from Volmer on RuO₂ (120.5 mV dec⁻¹) to Volmer-Heyrovsky on Ru/RuO₂ (55.0 mV dec⁻¹), suggesting that Ru can promote the H* combination step. H_{upd} measurement and the operando EIS test further verify that in addition to Pt single atoms, Ru can also enhance the H* adsorption to facilitate the H₂ generation. Moreover, the DFT calculations further suggests that Ru has a more optimized H* binding energy and lower H₂O adsorption ability than RuO₂.

It is worth noting that although both Pt and Ru are confirmed as the hydrogen-binding active sites, the former are more essential and indispensable considering the larger concentration of adsorbed H* and more remarkable HER activity enhancement caused by Pt single atoms. Therefore, we should conclude that all Pt, Ru and RuO₂ are active sites for activating the alkaline HER. Specifically, the water dissociation step is accelerated by RuO₂, meanwhile the followed H* combination step is primarily facilitated by Pt single atoms and also partially contributed by Ru, jointly leading to the enhanced HER activity on Pt-Ru/RuO₂.

Change: The SCN⁻ poisoning experiments on both Ru/RuO₂ and RuO₂ have been added in the Supplementary Figure 28. The related clarification has also been added from Line 303 to Line 304: “Besides, compared with the RuO₂, the obvious activity loss on Ru/RuO₂ indicates that Ru also

contributes to alkaline HER (Supplementary Fig 28).” Furthermore, the discussions of various active sites in Pt-Ru/RuO₂ have also been modified in the revised manuscript.

Supplementary Figure 28. Polarization curves of (a) Ru/RuO₂ and (b) RuO₂ before and after adding the poisoning SCN⁻.

5. There are some DFT calculations in this study, however, the author only calculated H* and H₂O* adsorption energy. The free energy diagram for whole HER should be provided.

Response: We thank the reviewer for providing this valuable suggestion. As suggested, we have added the free energy calculations for whole HER process (H₂O* dissociation and H* adsorption) in the revised manuscript. As demonstrated in the H₂O* dissociation free energy diagram in Supplementary Figure 39, Pt single atoms possess the largest energy barrier to break the molecule bond in H₂O. In addition, although Ru sites in metallic Ru and RuO₂ show similar energy barriers, considering that H₂O is prone to be adsorbed on RuO₂ (H₂O adsorption energy in Figure 4c), the water dissociation step is more possible and easily to be achieved by the RuO₂ in Pt-Ru/RuO₂. Corresponding H* adsorption free energy diagram in Figure 4d reveals that Pt sites deliver the most optimal value of 0.115 eV among all studied active sites, indicating that Pt single atoms are in favor of boosting the H* combination step. It is also noted that the H* free energy on Ru is closer to zero than RuO₂, suggesting that Ru also contributes to the H* adsorption and H₂ generation. These DFT calculation results are in agreement with the experimental results.

Change: Following the suggestion of the reviewer, we have calculated the free energy for whole HER process (H₂O* dissociation and H* adsorption) and added them in the Supplementary Figure 39 and Figure 4d, respectively. Related discussion has also been clarified from Line 406 to Line 416 in the revised manuscript: “The H₂O* dissociation free energy in Supplementary Fig. 39 further indicates that Pt single atoms is difficultly break the H₂O molecule due to the largest energy barrier. In addition, although Ru and RuO₂ have the similar energy barriers, considering the stronger ability of adsorbing H₂O on RuO₂, the water dissociation is more efficient and easier to be completed by RuO₂ rather than Ru. On the other hand, the H* adsorption free energy on all sites in Pt-Ru/RuO₂ were also calculated to examine their abilities for activating the followed H* combination step. As shown in Fig. 4d, compared to the other active sites, Pt sites deliver the optimal value of 0.115 eV, demonstrating that the incorporated Pt single atoms are favorable for boosting the H* combination step.⁶¹⁻⁶³ It is also noted that the H* adsorption free energy on Ru is

closer to zero than RuO_2 , which suggests that Ru also contributes to the H^* adsorption and H_2 generation.”. In addition, corresponding calculation methods for free energy have also been added in the Methods section.

Figure 4d. The H^* adsorption free energy values for Pt, Ru and RuO_2 sites in Pt-Ru/ RuO_2 .

6. In the operando Raman spectra, the author claimed they observe that the ν_3 peak proportions on both Pt-Ru/ RuO_2 and Ru/ RuO_2 display a dramatically faster downward trend than that of Pt/C with the increased HER potentials. However, in Figure 3a to 3c, we could not see this trend, the ν_3 peak is very weak in all of the samples.

Response: We thank the reviewer for raising this point. Indeed, as we mentioned in the manuscript, the broad Raman peak of interfacial water at around 3500 cm^{-1} is constituted by three kinds of coordinated water molecules on the catalyst surface (ν_1 at 3225 cm^{-1} , ν_2 at 3450 cm^{-1} and ν_3 at 3615 cm^{-1}), in which the ν_3 peak assigned to inactive dangling O-H bond only takes up a small portion. Generally, the higher the water dissociation ability of a catalyst, the smaller fraction the ν_3 peak. And the ν_3 peak proportion decreases further with the increase of the applied catalytic potentials, thus it is hard to directly distinguish the dynamic evolution trend of the ν_3 peak by visual inspection during the HER.

As shown in Supplementary Figure 33-35 and Supplementary Table 7, after carefully analyzing and fitting these broad peaks, the ν_3 peak proportions for all samples are elaborately evaluated, corresponding values are close to the reported values in the reported literatures [*Adv. Energy Mater.* **13**, 2203136 (2023); *Adv. Funct. Mater.* **33**, 2212321 (2023); *Adv. Funct. Mater.* **32**, 2109556 (2022)]. Meanwhile, it can be seen that the ν_3 peak fractions decrease with negatively increasing the applied potentials, in which Pt/C shows a relative slower downward trend compared with the Pt-Ru/ RuO_2 and Ru/ RuO_2 , indicating its unsatisfied water dissociation ability for alkaline HER.

7. In the operando XANES results, the author only measured one negative potential (-0.2 V). In fact, under such a negative overpotential, most metal oxides will be reduced, so this result cannot reflect the actual changes in active sites. The catalyst reported by the author exhibits high activity even at a very low negative potential (-0.018 V), therefore the author should test in-situ XANES near this potential in stages, rather than just -0.2 V.

Response: We thank the reviewer for providing this instructive comment. Following this suggestion, we have added the operando XANES measurements of Pt L_3 -edge and Ru K -edge at HER operating potential of -0.018 V (at which the current density is 10 mA cm^{-2}). As shown in Supplementary Figure 36a, once the potential of -0.018 V was imposed on Pt-Ru/RuO₂, the Pt L_3 -edge white line peak position displays an obvious negative shift, indicating that the Pt single atoms are reduced to lower valence states during HER. Meanwhile the operando Ru K -edge XANES in Supplementary Figure 36b shows that the Ru- K adsorption edge also shifts to lower energy at -0.018 V, suggesting the decreased oxidation states of Ru elements. Consequently, the operando XANES results confirm that both Pt and Ru elements in Pt-Ru/RuO₂ are active sites and participate in activating the alkaline HER.

Change: We have added the operando XANES spectra of Pt L_3 -edge and Ru K -edge at the HER operating potential of -0.018 V in the Supplementary Figure 36. Related clarification has also been added from Line 359 to Line 364 in the revised manuscript: “In order to further confirm that both Pt and Ru are active sites during HER, the marked lower HER potential of -0.018 V (at which the current density is 10 mA cm^{-2}) was applied on Pt-Ru/RuO₂ to conduct the operando Pt L_3 -edge and Ru K -edge XANES. Similarly, both white line peak position of Pt and adsorption edge of Ru shift to lower energy at -0.018 V (Supplementary Figure 36). Thus, the decrease in the valences states further verifies that both Pt and Ru elements in Pt-Ru/RuO₂ are functionalized as the active sites toward alkaline HER.”.

Supplementary Figure 36. The (a) Pt L_3 -edge and (b) Ru K -edge XANES spectra of Pt-Ru/RuO₂ measured in 1 M KOH solution under pristine, OCP and HER operating conditions of -0.018 V.

REVIEWER COMMENTS

Reviewer #1 (Remarks to the Author):

The authors performed necessary changes in the manuscript and addressed reviewer comments. The reviewer recommends no further revisions

Reviewer #2 (Remarks to the Author):

We appreciate the authors' revisions to the article based on our previous suggestions. However, since the novelty of the study is based on molecular-level insights into interfacial catalytic processes and dynamic changes of electrocatalysts, there are still some scientific issues should be clarified before its possible publication.

1. The authors claimed the water dissociation step is accelerated by RuO₂, meanwhile the followed H* combination step is primarily facilitated by Pt single atoms and also partially contributed by Ru, jointly leading to the enhanced HER activity on Pt-Ru/RuO₂. We agree with the H* part, however, the function of RuO₂ during water dissociation step still lack experimental proofs. First of all, many HER electrocatalysts based on Pt and Ru show fast water dissociation step without RuO₂. Secondly, even though the ν_3 peak proportions on both Pt-Ru/RuO₂ and Ru/RuO₂ display a dramatically faster downward trend than that of Pt/C with the increased HER potentials, which does not mean the water dissociation process is benefit from RuO₂. Except for RuO₂, Pt-Ru in Pt-Ru/RuO₂ is also quite different from that of Pt in Pt/C, the Pt-Ru could also facilitate water dissociation process. Finally, pure Ru and pure RuO₂ should also be compared to exclude the influence of synergistic effect between Ru/RuO₂.

2. The operando XAFS characterization in this study is not rigorous enough. They measured three edges for each element including pristine state and OCP, which means there is only one data under reaction condition. This is not accurate enough for getting any conclusion from operando experiments (Chem. Rev. 2021, 121, 2, 882–961). That's why we suggest the authors should test in-situ XANES near -0.018V in stages in our last review report, which means a series of data under reaction condition should be collected for comparison. However, the authors still just measures one data at -0.018V.

Reply to Reviewers on a point-by-point basis

Reviewer #1 (Remarks to the Author):

The authors performed necessary changes in the manuscript and addressed reviewer comments. The reviewer recommends no further revisions.

Response: We thank the reviewer for the positive conclusion for our revised manuscript and for recommending the acceptance of the manuscript.

Reviewer #2 (Remarks to the Author):

We appreciate the authors' revisions to the article based on our previous suggestions. However, since the novelty of the study is based on molecular-level insights into interfacial catalytic processes and dynamic changes of electrocatalysts, there are still some scientific issues should be clarified before its possible publication.

Response: We thank the reviewer for the positive assessment of our previous revision and we appreciate these useful comments to help us further improve the quality of the manuscript.

1. The authors claimed the water dissociation step is accelerated by RuO₂, meanwhile the followed H* combination step is primarily facilitated by Pt single atoms and also partially contributed by Ru, jointly leading to the enhanced HER activity on Pt-Ru/RuO₂. We agree with the H* part, however, the function of RuO₂ during water dissociation step still lack experimental proofs. First of all, many HER electrocatalysts based on Pt and Ru show fast water dissociation step without RuO₂. Secondly, even though the ν_3 peak proportions on both Pt-Ru/RuO₂ and Ru/RuO₂ display a dramatically faster downward trend than that of Pt/C with the increased HER potentials, which does not mean the water dissociation process is benefit from RuO₂. Except for RuO₂, Pt-Ru in Pt-Ru/RuO₂ is also quite different from that of Pt in Pt/C, the Pt-Ru could also facilitate water dissociation process. Finally, pure Ru and pure RuO₂ should also be compared to exclude the influence of synergistic effect between Ru/RuO₂.

Response: We thank the reviewer for raising this point. We will go through the various points raised by the reviewer point-by-point. We want to start with the role of Pt atoms: Figure 3d evidences that the decline rate of ν_3 peak proportion on Pt-Ru/RuO₂ (56.5%) is almost the same as that on Ru/RuO₂ (57.5%), suggesting the doped Pt single atoms have no contribution on water dissociation step. This is recognized in literatures that Pt atoms by themselves show a rather large energy barrier for activating and breaking water molecule [*Nat. Catal.* **4**, 711-718 (2021); *Nat.*

Energy **5**, 891-899 (2020); *Nat. Mater.* **5**, 909-913 (2006)], and is further verified by *operando* Raman of Pt/C and DFT in our presented work.

The role of Ru atoms. Although Ru performs better water activation ability than Pt, it cannot efficiently adsorb the water molecule and thus also has an insufficient water dissociation ability [*Energy Environ. Sci.* **14**, 5433-5443 (2021); *Angew. Chem. Int. Ed.* **61**, e202209486 (2022); *Adv. Mater.* **35**, 2301133 (2023)]. A documented strategy of designing active alkaline HER electrocatalysts is to support Pt/Ru on hydrous metal oxides [*Science* **334**, 1256-1260 (2011); *Angew. Chem. Int. Ed.* **59**, 14533-14540 (2020); *Nat. Commun.* **13**, 2024 (2022); *Nat. Commun.* **13**, 5382 (2022); *Nat. Mater.* **22**, 1022-1029 (2023)]. Therefore, RuO₂, as one of the state-of-art OER catalysts with an excellent water adsorption and dissociation ability, is adopted to efficiently break the water bond in this work.

In order to experimentally confirm this, we have further conducted the *operando* Raman characterizations of Ru and RuO₂ under HER operating conditions (Supplementary Figure S36-38). After carefully fitting the interfacial water peak in these spectra, we report in Supplementary Figure S39 that the ν_3 peak proportion at 3615 cm⁻¹ on RuO₂ alone displays a clearly faster downward trend in HER process than that on Ru alone, indicating its greater capability for cleaving the water bond. More importantly, pure RuO₂ exhibits a similar downward trend as the hetero-interfaced Ru/RuO₂ during more cathodic HER electrode potentials: this is a testament and validation to our hypothesis that the key water activation and dissociation sites on Ru/RuO₂ are located on the surface of RuO₂ rather than on the Ru phase, and this is consistent with our theoretical calculations. Moreover, it is calculated that the decline rate of ν_3 peak proportion on RuO₂ (52.4%) is slightly slower than that on Ru/RuO₂ (57.5%), which can be attributed to the electronic interaction between Ru and RuO₂ in Ru/RuO₂. As discovered in Supplementary Figure S42, Bader charge analysis reveals that electrons transfer from Ru to RuO₂ in Ru/RuO₂, which can modulate the electronic structure of RuO₂ to facilitate the catalytic ability [*Adv. Mater.* **35**, 2208821 (2023); *Nat. Commun.* **13**, 5448 (2022); *Angew. Chem. Int. Ed.* **61**, e202202519 (2022)].

Change: We have added the *operando* Raman spectra of Ru and RuO₂ under HER operating conditions and corresponding fitting results of the interfacial water peaks in Supplementary Figure S36-38. The ν_3 peaks proportions at nearly 3615 cm⁻¹ during HER on RuO₂, Ru and Ru/RuO₂ are also supplemented in Figure S39 and Supplementary Table 7. Related clarification has also been added from Line 349 to Line 361 in the revised manuscript: “As shown in Supplementary Fig. 36-38, the *operando* Raman tests were further conducted on Ru and RuO₂. After carefully fitting these spectra, it can be obviously observed that RuO₂ exhibits a faster downward trend of ν_3 peak proportion than that on Ru in HER process, confirming its greater capability for cleaving the water bond. Significantly, RuO₂ exhibits a similar downward trend as the Ru/RuO₂ during more cathodic HER potentials, which validates that RuO₂ is the main active site for water activation and dissociation step in Ru/RuO₂ instead of Ru. Furthermore, it is calculated that the decline rate of ν_3 peak proportion on RuO₂ (52.4%) is slightly slower than that on Ru/RuO₂ (57.5%), which may be attributed to the electronic interaction between Ru and RuO₂ in Ru/RuO₂ (Supplementary Fig. 39). Based on above investigations, it can be experimentally concluded that RuO₂ is primarily responsible for the water dissociation step while both Pt single atoms and Ru facilitate the followed hydrogen combination step, collectively leading to the remarkable alkaline HER activity of Pt-Ru/RuO₂, and this is further validated by DFT calculations.”.

Supplementary Figure 37. The fitting results of the interfacial water peak in the Raman spectra of RuO₂.

Supplementary Figure 38. The fitting results of the interfacial water peak in the Raman spectra of Ru.

Supplementary Figure 39. The proportions of ν_3 peaks at 3615 cm^{-1} during HER on RuO₂, Ru and Ru/RuO₂.

2. The operando XAFS characterization in this study is not rigorous enough. They measured three edges for each element including pristine state and OCP, which means there is only one data under reaction condition. This is not accurate enough for getting any conclusion from operando experiments (Chem. Rev. 2021, 121, 2, 882–961). That’s why we suggest the authors should test in-situ XANES near -0.018V in stages in our last review report, which means a series of data under reaction condition should be collected for comparison. However, the authors still just measure one data at -0.018V.

Response: We thank the reviewer for this important suggestion. While we, in principle, fully agree with the collection of ever more spectroscopic data for an ever better characterization of the catalysts under all possible conditions, we have to be also realistic and emphasize that the *operando* XANES tests are extremely time-consuming measurements conducted at synchrotron facilities not in a lab environment. This is why the expected delay time of 9-12 months from applying for additional own beamtime to performing the synchrotron experiments appears out of proportion.

However, following and in order to satisfy the reviewer’s suggestion, we have “borrowed” some extra beamtime from colleagues to perform the suggested measurements. We now present additional spectroscopic data measured at the Pt L_3 -edge and Ru K -edge of our sample at the pristine state, OCP, and HER operating potential of -0.018 V and -0.04 V, and the *operando* XANES measured at -0.04 V is added in Supplementary Figure 40.

The new data evidence that the white line of the Pt L_3 -edge displays a clearly lower energy shift when the applied potential was changed from OCP to -0.018 V (at which the current density is 10 mA cm⁻²), indicating that the chemical oxidation state of Pt single atoms reduced during cathodic HER. In addition, the valence states of Pt further decreased with the negative increase of potential to -0.040 V. Similarly, the energy position of the Ru K -edge XANES in Supplementary Figure 40b shows lower energy shifts from OCP to -0.018 V and further to -0.04 V, suggesting a continuous decrease in oxidation states of Ru element. Therefore, according to the potential-dependent *operando* XANES results tested at pristine state, OCP and two HER operating potentials (-0.018 V and -0.04 V), it is convincing now to get a solid conclusion that both Pt and Ru elements in the Pt-Ru/RuO₂ are active sites and efficiently activate the alkaline HER [*Nat. Commun.* **13**, 1143 (2022); *J. Am. Chem. Soc.* **143**, 17117-17127 (2021); *Nat. Catal.* **2**, 134-141 (2019)].

Change: We have added the *operando* XANES spectra of Pt L_3 -edge and Ru K -edge at the pristine, OCP and HER operating potentials of -0.018 V and -0.040 V in the Supplementary Figure 40. Related clarification has also been added from Line 373 to Line 380 in the revised manuscript: “In order to further confirm that both Pt and Ru are active sites during HER, the lower HER operating potentials of -0.018 V (at which the current density is 10 mA cm⁻²) and -0.040 V were applied on Pt-Ru/RuO₂ to conduct the *operando* Pt L_3 -edge and Ru K -edge XANES. Similarly, both white line peak position of Pt and adsorption edge of Ru continuously shift to lower energy with the negatively increased potentials, indicating their decreased valence states in the HER process (Supplementary Figure 40). Thus, the potential-dependent operando XANES results further verify that both Pt and Ru elements in Pt-Ru/RuO₂ are active sites toward the alkaline HER.”.

Supplementary Figure 40. The (a) Pt L_3 -edge and (b) Ru K -edge XANES spectra of Pt-Ru/RuO₂ measured in 1 M KOH solution under pristine, OCP and HER operating conditions of -0.018 V and -0.040 V.

REVIEWERS' COMMENTS

Reviewer #2 (Remarks to the Author):

All of the comments given have been appropriately addressed, and valuable improvements were introduced to the manuscript. This revised draft is now being recommended for publication.

Reply to Reviewers on a point-by-point basis

Reviewer #2 (Remarks to the Author):

All of the comments given have been appropriately addressed, and valuable improvements were introduced to the manuscript. This revised draft is now being recommended for publication.

Response: We thank the reviewer for the positive assessment of our revision and recommending the revised manuscript to be published in *Nature Communications*.